# SCNS: Continual Personalization of Diffusion Models via Submodular Concept Neuron Selection

Zijie Peng [* 1]   Enneng Yang [* 1 2]   Yifei Cheng [1]   Hongliang Yuan [3]   Fei Ma [2]   Xiaochun Cao [1]   Li Shen [1 2]

## Abstract

Custom diffusion models (CDMs) have demonstrated impressive success in visual personalization tasks by enabling the generation of user-specific concepts. However, existing CDMs typically assume that personalized concepts are static and rely on costly model merging or sequential updates that are prone to catastrophic forgetting as new concepts are introduced. To address these limitations, we propose a **S**ubmodular **C**oncept **N**euron **S**election method (SCNS), to solve CDMs with continual personalized concepts, which formulates continual personalization as a constrained submodular optimization problem to select a compact yet empirically effective set of concept-specific neurons under diminishing returns. SCNS combines a Facility Location-based coverage objective to suppress semantic redundancy, a Fisher-weighted risk proxy to protect previously learned concepts, and a cost-aware greedy rule to balance stability and plasticity with extreme sparsity. Extensive experiments demonstrate that SCNS achieves state-of-the-art performance in image alignment and anti-forgetting, while enabling fusion-free continual personalization by modifying only 0.41% of the total parameters for each concept on average. Our implementation is available at SCNS.

## 1. Introduction

Latent Diffusion Models (LDMs) have achieved remarkable success in large-scale text-to-image generation (Rombach et al., 2022; Lu et al., 2025). However, adapting these models to user-specific concepts, such as personalized pets or unique objects, remains a challenge, as such concepts cannot be reliably specified by text prompts alone. To address this, existing personalization methods fine-tune pretrained LDMs using a small set of reference images (Gal et al., 2023a; Ruiz et al., 2023; Voynov et al., 2023; Kumari et al., 2023; Zhang et al., 2025).

While early approaches focused on single-concept adaptation, extending personalization to multi-concept scenarios introduces significant challenges, including attribute entanglement, and concept interference (Gal et al., 2023b; Zhang et al., 2024; Ma et al., 2024; Jang et al., 2024). Existing techniques often assume a static setting, relying on the storage and post-hoc fusion of concept-specific weights (Kumari et al., 2023; Gu et al., 2023; Po et al., 2024). However, this assumption is impractical for real-world applications where concepts evolve incrementally. A more realistic paradigm is continual personalization, where concepts arrive sequentially (Yang et al., 2023; 2025a; Wang et al., 2024). Crucially, this strict replay-free constraint is driven not only by storage limits but also by data privacy concerns (Carlini et al., 2023). In many user-centric applications, retaining uploaded reference images for joint training poses significant security risks and violates data protection regulations. In this setting, standard continual learning methods attempt to mitigate catastrophic forgetting through consolidation losses, elastic weight aggregation, and distillation (Smith et al., 2023; Dong et al., 2024). Yet, they often struggle with the plasticity-stability trade-off, as cumulative regularization can overly constrain the model, hindering its ability to learn new concepts effectively (Smith et al., 2023).

To improve parameter efficiency, prior research has investigated the role of individual neurons in deep networks, aiming to identify representations crucial for specific tasks (Antverg & Belinkov, 2021; Wang et al., 2022; Durrani et al., 2020). Recently, this neuron-level analysis has been extended to continual diffusion personalization to guide fusion-free learning (Liao et al., 2025). However, these state-of-the-art methods typically rely on magnitude- or gradient-based heuristics to select neurons. Such approaches operate under an independence assumption, evaluating each parameter in isolation. Consequently, they over-

*Equal contribution [1]School of Cyber Science and Technology, Shenzhen Campus of Sun Yat-sen University, China [2]Guangdong Laboratory of Artificial Intelligence and Digital Economy (SZ), China [3]Xiaomi Corporation, China. Correspondence to: Li Shen <shenli6@mail.sysu.edu.cn>.

*Proceedings of the 43rd International Conference on Machine Learning*, Seoul, South Korea. PMLR 306, 2026. Copyright 2026 by the author(s).

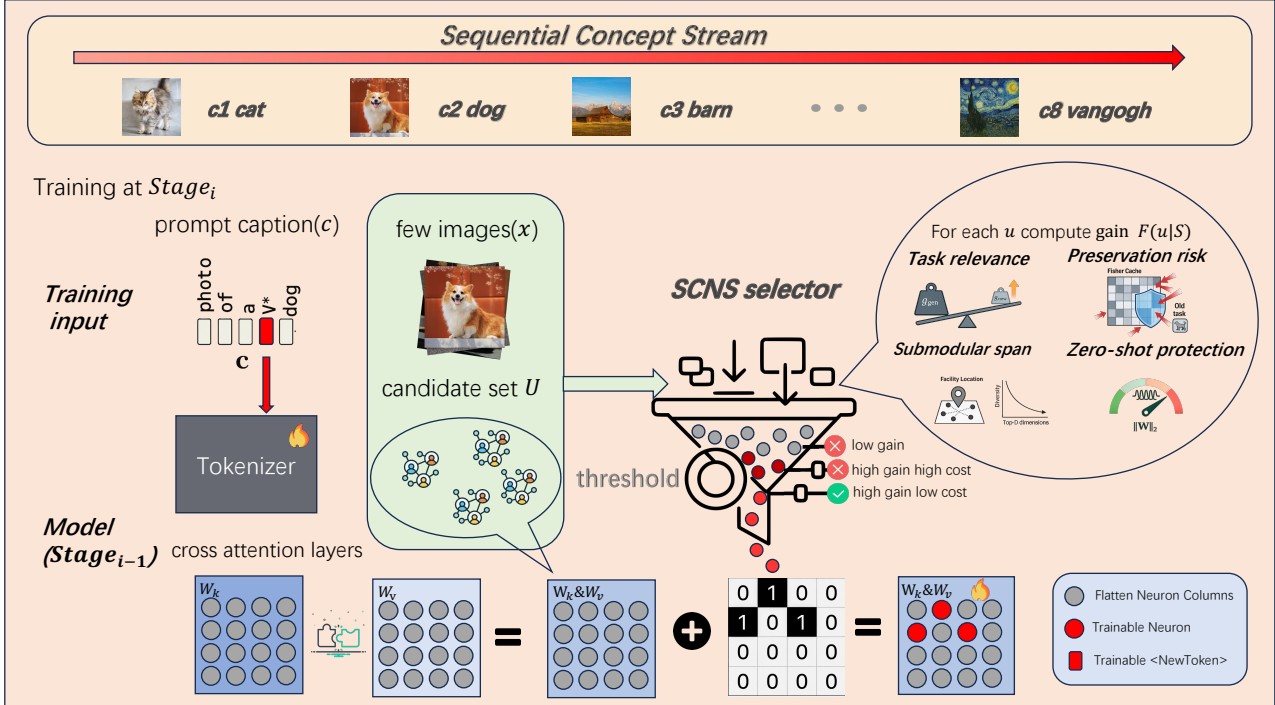

*Figure 1.* **Overview of the SCNS framework for continual personalization. (a) Sequential Concept Stream:** The system receives a continuous stream of personalized concepts (e.g., $c_1 \rightarrow c_8$) arriving sequentially. **(b) Training at Stage $i$:** Initialized with the model parameters from Stage $i-1$, the training process takes few-shot reference images and a prompt containing a new learnable token $V^*$ as input. While the tokenizer optimizes the embedding $V^*$, all column units in the cross-attention projection layers ($W_k, W_v$) constitute the Candidate Set $U$. **(c) SCNS Selector & Sparse Update:** The selector evaluates neurons via four objectives: Task Relevance, Preservation Risk, Submodular Span, and Zero-shot Protection. By optimizing Gain Density, the method applies a threshold-based filter to select a minimal sufficient subset, generating a sparse binary mask for parameter-efficient fine-tuning.

look semantic redundancy and collective coverage, often leading to parameter collisions and inefficient capacity expansion when adapting to semantically similar concepts.

To overcome these limitations, we formulate the parameter selection problem as a constrained submodular optimization task. Drawing on the theoretical foundations of submodularity (Edmonds, 2003) and the "Less is More" principle effectively applied in attribution tasks (Chen et al., 2024), we propose Submodular Concept Neuron Selection (SCNS). SCNS explicitly models the diminishing returns of adding parameters. By employing a Facility Location-based objective, SCNS selects a neuron subset that is compact yet empirically effective, maximizing semantic coverage while suppressing functional redundancy. To further ensure stability, we incorporate a Fisher-weighted second-order risk proxy, which identifies and preserves parameters crucial for historical tasks. Finally, by introducing a cost-aware greedy selection rule, SCNS automatically determines the optimal sparsity level for each concept, achieving a Pareto-optimal balance between stability and plasticity. Our focus is continual acquisition and retention: SCNS learns each arriving concept without replaying prior data, but does not explicitly address inference-time multi-concept composition, which

involves additional binding and layout-control challenges.

Through extensive experiments on a continual concept stream, we demonstrate that SCNS significantly outperforms state-of-the-art baselines. It achieves superior anti-forgetting and image alignment while updating only a minute fraction of the cross-attention parameters. The key contributions of this paper are as follows:

- We reformulate continual diffusion personalization as a constrained submodular subset selection problem, providing a principled mathematical foundation that moves beyond heuristic ranking methods.

- We introduce SCNS, a unified framework that jointly optimizes task relevance, Fisher-weighted retention risk, and semantic coverage, ensuring robust performance across complex sequential personalization tasks.

- We design a Facility Location-based objective that leverages the diminishing returns property to automatically suppress redundant neurons, enabling highly compact and parameter-efficient updates.

- We demonstrate that SCNS achieves state-of-the-art

performance in fusion-free continual learning, effectively mitigating catastrophic forgetting while maintaining high plasticity for new concepts.

## 2. Related Work

### 2.1. Diffusion Model Personalization

Diffusion model personalization enables pretrained models to synthesize user-defined concepts from sparse reference images. Early approaches like Textual Inversion (Gal et al., 2023a) optimize word embeddings within the text encoder, offering a lightweight solution but often limiting expressive capacity. Conversely, DreamBooth (Ruiz et al., 2023) finetunes the entire U-Net to achieve high fidelity, though this incurs significant storage overhead and risks language drift. To balance efficiency and fidelity, parameter-efficient finetuning (PEFT) methods, such as Custom Diffusion (Kumari et al., 2023) and LoRA (Hu et al., 2022), restrict updates to specific subsets of parameters, such as cross-attention projections or low-rank adapters. However, these paradigms are predominantly designed for single-concept adaptation, and directly applying them to a multi-concept scenario often leads to catastrophic forgetting.

### 2.2. Multi-Concept and Modular Customization

Extending personalization to multiple concepts introduces challenges in attribute entanglement and identity preservation. While joint training strategies (e.g., Custom Diffusion (Kumari et al., 2023)) can compose distinct concepts, they often suffer from "crosstalk" when subjects share semantic similarities. To address this, Modular Customization frameworks adapt models using separable weight components (Gu et al., 2023; Shah et al., 2024; Po et al., 2024; Yang et al., 2025b). Methods like Mix-of-Show (Gu et al., 2023) and Orthogonal Adaptation (Po et al., 2024) independently train task-specific adapters and merge them during inference. However, post-hoc fusion often requires computationally expensive optimization or degrades identity alignment as the number of merged concepts increases.

### 2.3. Continual Learning in Generative Models

Continual Personalization presents a stricter paradigm where concepts arrive sequentially. To mitigate catastrophic forgetting, regularization-based methods such as C-LoRA (Smith et al., 2023) and CIDM (Dong et al., 2024) penalize weight updates in sensitive regions. Recently, Concept Neuron Selection (CNS) (Liao et al., 2025) proposed updating only a sparse subset of concept-specific neurons in cross-attention layers. While effective for fusion-free learning, CNS relies on magnitude-based heuristics, which treat parameters independently and may overlook the collective redundancy within the selected subset.

### 2.4. Submodular Optimization in Deep Learning

Submodular optimization provides a rigorous framework for selecting informative subsets from high-dimensional data, characterized by the property of *diminishing returns* (Krause & Golovin, 2014; Edmonds, 2003; Sviridenko, 2004). In computer vision, this paradigm has been effectively applied to data subset selection and model interpretability. For instance, the SMDL framework reformulates image attribution as a submodular selection task to identify compact yet empirically effective regions for model decisions (Chen et al., 2024). Despite these advancements in data and feature selection, the application of submodular optimization to the *parameter space* of diffusion models remains an unexplored avenue, specifically for identifying non-redundant, task-specific neurons.

## 3. Preliminaries

### 3.1. Latent Diffusion Models (LDMs)

LDMs (Rombach et al., 2022) perform the denoising process in a compressed latent space to improve computational efficiency. Given an image $x$, an encoder $\mathcal{E}(\cdot)$ maps it to a latent representation $z = \mathcal{E}(x)$. The forward process adds Gaussian noise $\epsilon$ to $z$, producing a noisy latent $z_t$ at timestep $t$. The model learns a denoising network $\epsilon_\theta$ to predict the noise conditioned on textual information $c$ by minimizing

$$\mathcal{L}_{\text{LDM}} = \mathbb{E}_{z,c,\epsilon,t}\left[ w_t \left\| \epsilon - \epsilon_\theta(z_t, c, t) \right\|_2^2 \right], \qquad (1)$$

where $w_t$ is a noise-schedule-dependent weighting factor.

### 3.2. Cross-Attention Mechanism

In text-to-image LDMs, textual conditions are injected via cross-attention layers, which align latent image features with semantic content from text prompts (Rombach et al., 2022). Let $c \in \mathbb{R}^{s \times d}$ represent the text features (produced by a text encoder) and $f \in \mathbb{R}^{(h \times w) \times d}$ represent the latent image features. The cross-attention operation is defined as:

$$\text{Attention}(Q, K, V) = \text{softmax}\left( \frac{QK^\top}{\sqrt{d'}} \right) V,$$

where $Q = fW_q$, $K = cW_k$, $V = cW_v$ and $d'$ is the key/query dimension. Here, $W_q, W_k, W_v$ are learnable projection matrices. Crucially, as the text features $c$ are only input to $W_k$ and $W_v$, these two matrices govern the mapping from text to the visual domain. Following (Kumari et al., 2023), our method focuses on the column-wise adaptation of $W_k$ and $W_v$ to achieve parameter-efficient personalization.

### 3.3. Submodularity and Marginal Gain

Submodularity is a fundamental property of set functions defined on finite ground sets and is commonly regarded as a

discrete analogue of concavity. Let $V$ denote a finite ground set and $f : 2^V \to \mathbb{R}$ be a set function.

**Definition 3.1** (Submodular Function)**.** The function $f$ is submodular if, for any $A \subseteq B \subseteq V$ and any element $e \in V \setminus B$, the following inequality holds (Edmonds, 2003):

$$f(A \cup \{e\}) - f(A) \ge f(B \cup \{e\}) - f(B), \quad (2)$$

**Definition 3.2** (Marginal Gain)**.** The marginal gain of an element $e$ with respect to a set $S$ is formally defined in the literature as (Krause & Golovin, 2014):

$$\Delta(e \mid S) = f(S \cup \{e\}) - f(S). \quad (3)$$

While maximizing a submodular function is NP-hard, greedy algorithms provide strong theoretical guarantees. For cardinality constraints ($|S| \le k$), the standard greedy algorithm achieves a $(1 - 1/e)$ approximation (Nemhauser et al., 1978). Critically, for budgeted maximization where elements have heterogeneous costs, a density-based greedy strategy variant also maintains comparable constant-factor approximation guarantees (Lin & Bilmes, 2010).

# 4. Method

## 4.1. Problem Formulation

We address the problem of continual diffusion personalization, where personalized concepts arrive sequentially as a stream $\mathcal{C} = (c_1, \ldots, c_T)$. At stage $i$, the model observes a few-shot dataset $D_i = \{(x, p)\}$ containing reference images $x$ and prompts $p$ with concept tokens, without access to prior datasets $\{D_1, \ldots, D_{i-1}\}$. Let $\theta_{i-1}$ denote the model parameters after learning concepts up to $i-1$. The **objective** is to learn $\theta_i$ that accurately generates $c_i$ while minimizing the forgetting of prior concepts and preserving the zero-shot capabilities of the pretrained model. We optimize the standard LDM objective (Eq. 1) but restrict updates to a sparse subset of parameters to ensure efficiency.

Specifically, we focus on the *column-wise sparse* adaptation of the cross-attention key and value projection matrices, $W_k^{(\ell)}, W_v^{(\ell)} \in \mathbb{R}^{d \times d}$, in layer $\ell$. We define an adaptation unit ("neuron") as a column index $j \in \{1, \ldots, d\}$ in layer $\ell$, denoted by the tuple $u = (\ell, j)$. The set of all candidate units is $U = \{(\ell, j) \mid \ell \in [L], j \in [d]\}$. For each unit $u$, we define its gradient signature as the concatenation of the corresponding column gradients:

$$g(u) = \left[ \nabla W_k^{(\ell)}[:,j]; \nabla W_v^{(\ell)}[:,j] \right] \in \mathbb{R}^{2d}. \quad (4)$$

At stage $i$, our method selects a subset $S_i \subseteq U$ and applies a binary gradient mask, updating only the units in $S_i$ on $D_i$ while freezing the rest. This strategy yields a fusion-free, parameter-efficient continual learning process.

## 4.2. SCNS: Constrained Submodular Subset Selection

Given the candidate universe $U$, the core challenge is to identify an optimal subset $S \subseteq U$ for each concept that balances plasticity and stability. We formulate this as a constrained combinatorial optimization problem. Our goal is to select a subset that maximizes task relevance and semantic coverage while simultaneously minimizing both redundancy and interference with historical knowledge.

Let $F : 2^U \to \mathbb{R}$ be the selection objective function. We decompose $F$ into a *modular* term $s_{\text{lin}}(u)$ that captures the individual utility of each unit, and a *submodular* term $s_{\text{span}}(S)$ that enforces collective semantic coverage. SCNS solves the following maximization problem:

$$\max_{S \subseteq U} F(S) = \sum_{u \in S} s_{\text{lin}}(u) + \lambda_{\text{span}} \, s_{\text{span}}(S), \quad (5)$$

where $\lambda_{\text{span}}$ is a weighted hyperparameter, $s_{\text{lin}}(u)$ aggregates task relevance, preservation risk, and zero-shot safety, while the submodular term $s_{\text{span}}(S)$ models the diminishing returns of adding semantically similar neurons. This decomposition allows us to leverage greedy algorithms with theoretically bounded approximation errors.

**Modular Score Aggregation.** To ensure the mathematical consistency of the submodular framework, we formulate all individual utility scores as non-negative values. Specifically, the preservation term (derived from negative risk) is transformed via an exponential map, $s_{\text{pres}}(u) = \exp(-\text{LSE}(\mathcal{R}_p(u)))$, to map high-risk units to low (but positive) utility. LSE() and $\mathcal{R}_p(u)$ are introduced in Section 4.3. The unified modular score $s_{\text{lin}}(u)$ is then formally constructed and defined as a weighted sum:

$$s_{\text{lin}}(u) = \lambda_{\text{rel}} s_{\text{rel}}(u) + \lambda_{\text{pres}} s_{\text{pres}}(u) + \lambda_{\text{zero}} s_{\text{zero}}(u), \quad (6)$$

where $\lambda$ are non-negative balancing coefficients. This construction guarantees that the modular component of our objective is monotone and non-negative, satisfying the requirements for submodular maximization. Full pseudocode of the algorithm is given in Algorithm 1.

## 4.3. Scoring Components

### 4.3.1. TASK RELEVANCE

We first quantify the responsiveness of each unit to the current concept $c_i$. To distinguish concept-specific neurons from globally active ones (which may encode general syntax), we compare the gradient magnitude on the current dataset $D_i$ against a general response baseline computed on a fixed set of diverse universal semantic anchors. The anchor set is not intended to cover or enumerate future personalized concepts. Instead, it serves as a general-response baseline for identifying broadly activated "generalist" units.

**Algorithm 1** SCNS: Submodular Concept Neuron Selection

---

**Require:** Current concept dataset $D_i$; anchor prompt set $\mathcal{A}$; candidate set $U$ over columns of $\{W_k^{(\ell)}, W_v^{(\ell)}\}$; Fisher caches $\{f_p(\cdot)\}_{p<i}$; semantic dimension set $\mathcal{D}$; hyperparameters $\lambda_{\text{rel}}, \lambda_{\text{pres}}, \lambda_{\text{zero}}, \lambda_{\text{span}}, \alpha, \kappa, \gamma, c_0, \alpha_{\text{ov}}$; threshold $\tau$.

**Ensure:** Selected update subset $S_i \subseteq U$.
1: Compute per-unit gradients $g_{\text{new}}(u)$ on $D_i$ for all $u \in U$.
2: Compute general-response gradients $g_{\text{gen}}(u)$ on anchors $\mathcal{A}$ for all $u \in U$.
3: Compute $s_{\text{rel}}(u)$, and $s_{\text{pres}}(u)$ using Fisher caches and LogSumExp aggregation.
4: Compute $s_{\text{zero}}(u)$ and assemble $s_{\text{lin}}(u)$.
5: Initialize $S \leftarrow \emptyset$, cost $\leftarrow 0$.
6: **while** true **do**
7:    For each $u \in U \setminus S$, evaluate marginal gain $\Delta F(u \mid S)$ and density $\text{density}(u)$.
8:    $u^\star \leftarrow \arg\max_{u \in U \setminus S} \text{density}(u)$.
9:    **if** $\text{density}(u^\star) < \tau$ **then**
10:      **break** {cost-aware adaptive stopping}
11:   **end if**
12:   **if** $\text{cost} + \text{Cost}(u^\star) > B$ **then**
13:      **break** {optional budget constraint}
14:   **end if**
15:   $S \leftarrow S \cup \{u^\star\}$, cost $\leftarrow$ cost $+ \text{Cost}(u^\star)$.
16: **end while**
17: **return** $S_i \leftarrow S$.

---

Thus, SCNS does not require the target concept itself to appear in the anchor distribution.

The relevance score is defined as a signal-to-noise ratio:

$$s_{\text{rel}}(u) = \frac{\|g_{\text{new}}(u)\|_2}{\|g_{\text{gen}}(u)\|_2 + \varepsilon}, \qquad (7)$$

where $g_{\text{new}}(u)$ and $g_{\text{gen}}(u)$ are the gradient signatures for the current concept and the general anchors, respectively. $\varepsilon$ is a tiny constant to prevent division by zero. A high $s_{\text{rel}}(u)$ indicates that unit $u$ is specifically sensitive to the unique features of the new concept.

### 4.3.2. FISHER-WEIGHTED PRESERVATION

To mitigate catastrophic forgetting, we estimate the interference risk of updating unit $u$ with respect to previously learned concepts. For each past stage $p < i$, we maintain a diagonal Fisher information approximation $f_p(u) \approx \mathbb{E}[g_p(u)^2]$, reflecting the unit's importance for task $p$ (Kirkpatrick et al., 2017). Although this approximation cannot capture cross-column covariance, it provides a conservative and efficient signal for identifying units whose perturbation is likely to harm previously acquired concepts. From a Bayesian perspective, this quantity reflects the precision of the parameter posterior, indicating how critical the unit is for preserving prior knowledge (MacKay, 2003; Ritter et al., 2018).

We formulate the preservation score $s_{\text{pres}}(u)$ as the negative aggregation of historical retention risks. The risk for a past task $p$ jointly combines a first-order directional conflict term and a weighted second-order Fisher penalty:

$$\mathcal{R}_p(u) = \max\left(0, -\langle g_{\text{new}}(u), g_{\text{EMA}}^{(p)}(u)\rangle + \alpha f_p(u)\|g_{\text{new}}(u)\|_2^2\right), \qquad (8)$$

where $g_{\text{EMA}}^{(p)}$ tracks the smoothed gradient direction of task $p$. To enforce a conservative safety margin, we aggregate these risks via a smooth LogSumExp (LSE) operator. As mentioned in Eq. 6, this aggregated risk is then inverted to form the positive utility $s_{\text{pres}}(u)$. Detailed analysis of the Fisher proxy and risk formulation is provided in Appendix B.1.

### 4.3.3. SUBMODULAR SEMANTIC SPAN

To suppress semantic redundancy, we model neuron selection as a feature coverage problem, inspired by submodular approaches for document summarization (Lin & Bilmes, 2011). We employ a **Facility Location** function to identify a subset of neurons that maximally covers the salient semantic features of the target concept.

Let $E \in \mathbb{R}^D$ denote the text embedding of the concept token. We explicitly define the semantic universe $\mathcal{D}$ by selecting the indices of the **top-$K$ dimensions with the largest absolute magnitudes** in $E$. The span score is formulated as:

$$s_{\text{span}}(S) = \sum_{d \in \mathcal{D}} \max_{u \in S} \text{Sim}(u, d), \qquad (9)$$

where $\text{Sim}(u, d) = |\hat{w}(u)_d| \cdot |E_d|$ measures alignment intensity. Here, $\hat{w}(u)$ is the $L_2$-normalized concatenation of the pre-update key and value columns ($[W_k(u); W_v(u)]$). This normalization ensures robustness to cross-layer scaling, while absolute values capture the magnitude of responsiveness regardless of direction.

As a monotone submodular function, $s_{\text{span}}$ exhibits the property of **diminishing returns**: adding a neuron that covers semantic dimensions already spanned by existing members of $S$ yields marginal gain. This effectively penalizes functional redundancy, steering the optimization toward a diverse set of units that cover the concept's feature space efficiently (Krause & Golovin, 2014).

### 4.3.4. ZERO-SHOT PROTECTION

To preserve the pretrained model's generalization ability, we prioritize updates to parameters that are less structurally significant. Units with large weight norms typically encode robust, global semantic structures; modifying them risks degrading zero-shot performance. Instead of a hard penalty, we formulate a positive safety score that encourages the selection of units with smaller magnitudes:

$$s_{\text{zero}}(u) = \frac{\gamma}{\|W_k(u)\|_2 + \|W_v(u)\|_2}, \qquad (10)$$

where $\gamma$ is a small scaling factor.

*Table 1.* Quantitative comparison of image quality and subject fidelity after learning all concepts. Higher is better for all metrics.

| Method | CLIP-T ↑ | CLIP-I ↑ | DINO-I ↑ |
|---|---|---|---|
| Textual Inversion | 0.2781 | 0.7731 | 0.6162 |
| Custom Diffusion | 0.3024 | 0.7568 | 0.5648 |
| Mix-of-Show | 0.2980 | 0.7603 | 0.5997 |
| Orthogonal Adaptation | 0.2922 | 0.7207 | 0.5635 |
| Continual Diffusion | 0.2820 | 0.7297 | 0.5677 |
| CIDM | 0.3029 | 0.7362 | 0.5255 |
| CNS | 0.2875 | 0.7478 | 0.5837 |
| **SCNS (Ours)** | **0.3041** | **0.7854** | **0.6216** |

### 4.4. Greedy Selection with Cost-aware Stopping

To solve the optimization problem in Eq. 5, we employ a cost-aware greedy algorithm that maximizes the **Gain Density**—the ratio of marginal gain to cost. At each step $t$, we select the unit $u^*$ that maximizes:

$$\rho(u \mid S_{t-1}) = \frac{F(S_{t-1} \cup \{u\}) - F(S_{t-1})}{\text{Cost}(u)}, \qquad (11)$$

where $\text{Cost}(u) = c_0 + \alpha_{\text{ov}} \cdot \text{Overlap}(u)$ and the constant $c_0$ serves as a baseline overhead for selecting any unit, ensuring a stable gain-density scale regardless of prior task history. $\text{Overlap}(u)$ counts the frequency with which unit $u$ has been updated in previous tasks. This overlap penalty actively steers the greedy selection toward "fresh" parameter subspaces to reduce inter-task interference.

**Adaptive Stopping and Theoretical Properties.** As a consequence of submodularity, the marginal gain density $\rho$ decreases monotonically as the selected subset grows. We therefore adopt a threshold-based stopping rule that terminates selection once $\rho(u^*) < \tau$, yielding a subset that is compact yet empirically effective for representing the current concept without imposing a fixed sparsity budget. From an optimization perspective, this criterion can be interpreted as a Lagrangian relaxation of a cost-constrained submodular maximization problem, where density-based greedy selection admits constant-factor approximation guarantees. This formulation places the solution on the utility–cost Pareto frontier, ensuring that each updated parameter contributes a marginal efficiency exceeding $\tau$ and allowing model capacity to adapt automatically to concept complexity (Please refer to Appendix B.2 for details).

## 5. Experiments

### 5.1. Experimental Setup

**Benchmark and Protocol.** We evaluate SCNS on a continual personalization benchmark consisting of a sequence of $T = 8$ diverse concepts (animals, objects, and styles), sourced from Custom Diffusion and DreamBooth datasets. Each concept is learned from a few-shot set of 5–6 images. We employ Stable Diffusion v1.5 as the backbone.

Consistent with our method, we fine-tune the selected cross-attention key/value projections and new concept embeddings for 500 steps with a learning rate of $1 \times 10^{-5}$. Full details and hyperparameters are provided in **Appendix C.1**.

**Baselines.** We compare SCNS against two categories of approaches: (1) **Merge-based methods**, including Textual Inversion (TI) (Gal et al., 2023a), Custom Diffusion (CD) (Kumari et al., 2023), Mix-of-Show (Gu et al., 2023), and Orthogonal Adaptation (Po et al., 2024). These methods train concepts independently and fuse weights post-hoc. (2) **Continual Learning methods**, including C-LoRA (Smith et al., 2023), CIDM (Dong et al., 2024), and CNS (Liao et al., 2025), which learn concepts sequentially without accessing past data. All baselines are implemented using official configurations (see **Appendix C.5**).

**Evaluation metrics.** For the quantitative evaluation, we follow existing protocols (Kumari et al., 2023) to use CLIP-based scores, DINO-based identity scores, and Task Forgetting Rate (TFR) as metrics. Specifically, for **Text-Alignment (CLIP-T)** and **Image-Alignment (CLIP-I)**, we utilize the respective encoders of CLIP (Radford et al., 2021) to compute the feature similarity between the synthesized image and the target prompt or the reference concept image. For **Identity Preservation (DINO-I)**, we use the DINO (Caron et al., 2021) vision transformer to evaluate instance-level feature consistency, which offers a more robust assessment of subject fidelity than CLIP alone. Finally, to measure stability, we report the **Task Forgetting Rate (TFR)**. It is calculated as $(S_{\text{init}} - S_{\text{final}})/S_{\text{init}} \times 100\%$, where $S_{\text{init}}$ is the metric score (CLIP-I or DINO-I) evaluated immediately after a concept's initial learning stage, and $S_{\text{final}}$ is its score at the final stage of the continuum.

### 5.2. Qualitative Comparisons

To ensure a comprehensive assessment, we employ a diverse set of prompts—ranging from object re-contextualization to stylistic rendering to challenge the models generalization capabilities (see **Appendix C.2**). Fig. 6 (in Appendix A) visualizes the generation results across the 8-concept stream. As illustrated, **SCNS** outperforms existing methods in maintaining subject fidelity across the entire concept stream.

Textual Inversion often fails to capture complex visual characteristics (e.g., the mechanical details of `robot`) due to its limited expressive capacity. Custom Diffusion and LoRA-based merging methods, meanwhile, increasingly suffer from *concept vanishing* or attribute leakage as the sequence grows; for example, early concepts like `dog` become generic or corrupted in later stages. For sequential baselines such as Continual Diffusion and CIDM, we observe a different failure mode caused by over-regularization: in `teddybear` and `cartoon`, the outputs deviate visually from the ref-

*Table 2.* Sequential (continual) personalization performance comparison after learning the full stream of eight concepts. Metrics marked with ↑ (CLIP-T/CLIP-I/DINO-I) indicate that higher scores are better, while ↓ denotes that lower Task Forgetting Rates (TFR) are better. (Note. Custom Diffusion is evaluated under a sequential protocol, fine-tuning text embeddings and cross-attention layers per concept.)

| METHOD | CLIP-T ↑ | CLIP-I ↑ | DINO-I ↑ | TFR-CLIP ↓ | TFR-DINO ↓ |
|---|---|---|---|---|---|
| *External continual baselines* | | | | | |
| CUSTOM DIFFUSION (SEQ) | 0.2886 | 0.7324 | 0.5085 | 5.3372 | 11.0478 |
| CONTINUAL DIFFUSION | 0.2820 | 0.7297 | 0.5677 | 0.4782 | 1.1874 |
| CIDM | 0.3029 | 0.7362 | 0.5255 | 0.9708 | 1.7437 |
| CNS | 0.2875 | 0.7478 | 0.5873 | 1.6671 | 3.0564 |
| *SCNS and ablations* | | | | | |
| SCNS W/O NEURON SELECTION | 0.2859 | 0.7403 | 0.5229 | 11.092 | 26.4386 |
| SCNS W/ RANDOM SELECTION | 0.2908 | 0.7635 | 0.5873 | 4.3302 | 9.0304 |
| SCNS (SCORES ONLY; W/O COST-AWARE) | 0.2830 | 0.7839 | 0.6212 | 0.8116 | 0.5954 |
| **SCNS (FULL; SCORES + COST-AWARE)** | **0.3041** | **0.7854** | **0.6216** | **0.1183** | **0.3439** |

*Figure 2.* **CLIP-I decline curves under different continual personalization strategies.** Each line tracks the identity similarity (CLIP-I) of a concept as new concepts are sequentially learned. Dense fine-tuning and random sparse updates suffer from rapid forgetting, while score-only selection partially mitigates degradation but still exhibits noticeable interference. In contrast, **SCNS (full)** maintains consistently higher CLIP-I scores with significantly flatter decline curves through submodular coverage and cost-aware allocation.

erences and miss fine-grained details. We attribute this to strong consolidation terms (e.g., EWC or self-regularization) that reduce forgetting but also hinder plasticity, limiting adaptation to new, complex concepts. In contrast, SCNS better balances stability and plasticity, enabling robust continual personalization without post-hoc fusion.

## 5.3. Quantitative Comparisons

We conduct a comprehensive quantitative evaluation focusing on two key dimensions: final generation fidelity and continual learning stability. All reported results are aggregated over a rigorous evaluation protocol using 20 diverse prompts per concept. Specifically, the assessment involves generating **1,600 images** for the final quality comparison and **7,200 images** for the cumulative forgetting analysis to ensure statistical reliability (Please refer to Appendix C.1).

### 5.3.1. IMAGE QUALITY AND SUBJECT FIDELITY

As reported in Tab. 1, SCNS consistently achieves the best performance across all three metrics. In particular, SCNS attains higher CLIP-I and DINO-I scores than both merging-based and continual learning baselines, indicating superior identity preservation after long concept sequences. These

results suggest that SCNS effectively balances expressive capacity and parameter isolation, avoiding the identity degradation commonly observed in LoRA-based fusion and over-regularization in continual methods. Table 1 reports the average performance across all concepts after the full continual learning stream. To provide a finer-grained view of SCNS, we additionally report stage-wise concept-level results in Appendix A.

### 5.3.2. CATASTROPHIC FORGETTING ANALYSIS

As shown in Tab. 2 and Fig. 2, SCNS exhibits a substantially lower task forgetting rate compared to other continual personalization baselines and ablation studies. Notably, SCNS maintains a significantly flatter degradation curve, indicating that identity fidelity for earlier concepts is preserved more effectively throughout the learning process. The significantly flatter degradation curve empirically validates our design: the Fisher-weighted constraint effectively shields sensitive parameters, while the submodular span term prevents new concepts from destructively overwriting historical representations, thereby achieving a Pareto-optimal balance between plasticity and stability.

*Table 3.* Component-wise ablation of SCNS. Each variant removes one scoring component by setting the corresponding weight to zero. Higher is better for CLIP-T/CLIP-I/DINO-I, and lower is better for TFR.

| Variant | CLIP-T ↑ | CLIP-I ↑ | DINO-I ↑ | TFR-CLIP ↓ | TFR-DINO ↓ |
|---|---|---|---|---|---|
| w/o task relevance | 0.3026 | 0.7684 | 0.5893 | 3.7251 | 5.3487 |
| w/o preservation | 0.2976 | 0.7774 | 0.6099 | 1.4342 | 2.4477 |
| w/o submodular span | 0.2924 | 0.7711 | 0.6057 | 0.9669 | 1.9114 |
| w/o zero-shot protection | 0.2952 | 0.7792 | 0.6196 | 0.3780 | 0.5933 |
| **SCNS full** | **0.3041** | **0.7854** | **0.6216** | **0.1183** | **0.3439** |

## 5.4. Ablation Studies

We conduct ablations to isolate the contribution of each component and validate the design choices. Quantitative results are summarized in Tab. 2, and the corresponding score-decline curves are provided in Fig. 2.

**Dense cross-attention finetuning (w/o sparsity).** We fine-tune *all* columns in cross-attention layers. This dense adaptation leads to pronounced catastrophic forgetting: early concepts exhibit sharp degradation in CLIP-I/DINO-I shortly after learning subsequent concepts. This empirically confirms the necessity of sparse, concept-specific parameter-level updates for continual personalization.

**Random selection at matched sparsity.** We randomly select the same number of columns as SCNS and apply the identical training procedure. Random selection significantly degrades identity alignment and concept fidelity, indicating that *structured selection* is crucial to capture concept-relevant semantics rather than merely enforcing sparsity.

**Fixed-budget Top-$K$ without cost-aware stopping.** We disable the density-based greedy rule and instead select a fixed top-$K\%$ of units based on the modular score. This variant essentially functions as an enhanced gradient magnitude-based heuristic, but operates under a rigid capacity budget rather than an adaptive utility threshold. While this approach captures the current concept, it suffers from significantly higher interference on historical tasks. In contrast, our adaptive stopping strategy identifies a *compact yet empirically effective* neuron columns subset, effectively preventing over-allocation and reserving valuable parameter space for accommodating future concepts in the stream.

**Component-wise ablation.** To isolate the contribution of each scoring term, we remove one component at a time by setting its weight to zero. As shown in Tab. 3, each term contributes complementary benefits.

## 5.5. Theory-driven Analysis

To align the theoretical design of SCNS with its empirical behavior, we analyze the learned update subsets along a causal progression. We first show that diminishing returns

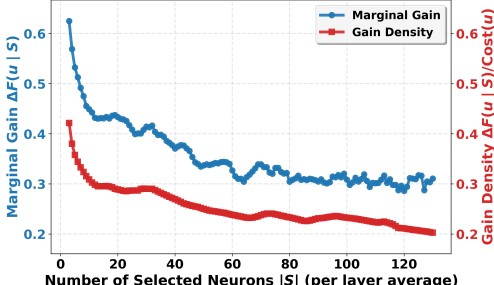

*Figure 3.* Greedy selection of SCNS illustrating diminishing returns. We plot the marginal gain $\Delta F(u \mid S)$ and gain density $\Delta F(u \mid S)/\text{Cost}(u)$ as functions of the selected set size $|S|$.

in the span score, together with cost-aware selection, yield compact yet sufficient parameter subsets. This property naturally leads to reduced redundancy and lower cross-concept overlap, improving robustness to interference. When the concept stream is extended, we further demonstrate that methods lacking these properties suffer from accelerated forgetting and capacity exhaustion, whereas SCNS maintains stability under longer concept sequences.

**Subset Sufficiency from Diminishing Returns and Density Thresholding.** SCNS is grounded in the diminishing-returns property of submodular objectives, which is empirically reflected by the decay curves in Fig. 3. As marginal gains decrease with growing subset size, the gain density provides a principled criterion for determining when additional parameters become inefficient. Rather than relying on heuristic geometric inflection points, the density-based stopping rule selects parameters as long as the utility-to-cost ratio exceeds the threshold $\tau$, ensuring that the resulting subset is both compact and sufficient. This mechanism allows selection to extend into the long tail of moderate-gain neurons when required, enabling the model to capture fine-grained semantic attributes beyond dominant features and preventing under-representation of complex concepts.

**Redundancy Suppression and Parameter Separation.** Given a sufficiently expressive subset for each concept, an immediate consequence is reduced competition for shared parameters across tasks. We quantify this effect by measuring the pairwise overlap of selected neuron subsets using the Jaccard similarity and visualize the resulting collision matrices in Fig. 4. Score-based sparse selection exhibits uniformly high overlap across concepts (0.35–0.40), indicating systematic redundancy that is largely insensitive to semantic relatedness. In contrast, SCNS achieves substantially lower overlap, with most off-diagonal entries below 0.10, reflecting effective separation of parameter subspaces. Limited sharing emerges only where semantically plausible, suggesting that cost-aware selection suppresses redundancy without enforcing rigid orthogonality.

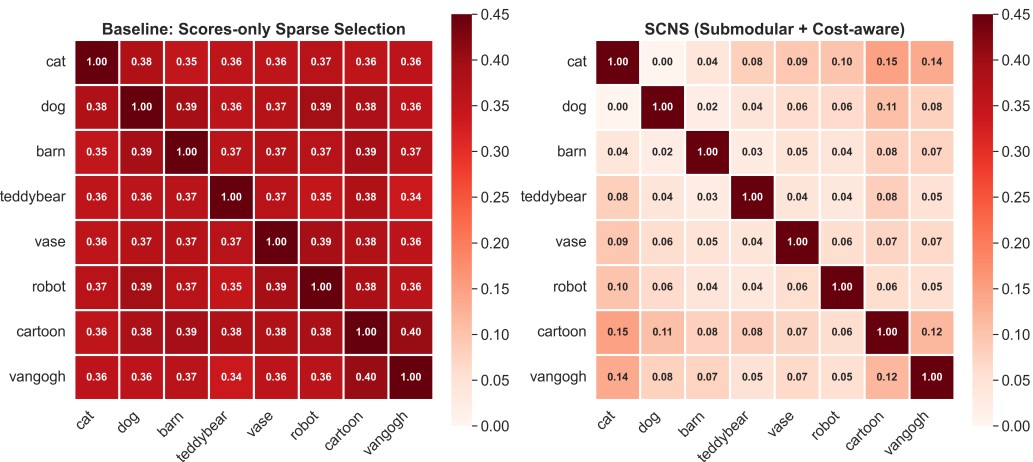

*Figure 4.* **Parameter collision analysis via neuron overlap.** We visualize the Jaccard similarity between neuron subsets selected for different concepts. **Left:** score-based sparse selection without cost-aware **Right:** SCNS (full)

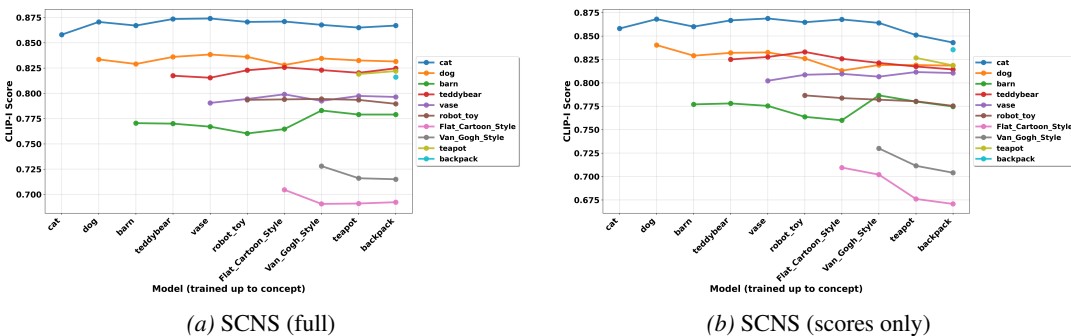

*(a)* SCNS (full)   *(b)* SCNS (scores only)

*Figure 5.* CLIP-I decline curves under 10 concepts. SCNS (full) exhibits smoother degradation than SCNS (scores only).

**Long-Stream Stability under Extended Concept Sequences.** The implications of redundancy accumulation become most evident when the concept stream is extended. As shown in Fig. 5, score-only selection without cost regularization repeatedly allocates capacity to a small set of high-gain neurons, leading to rapid saturation and insufficient parameter space for later concepts such as teapot and backpack. This saturation exacerbates interference, resulting in pronounced degradation of earlier styles. In contrast, SCNS leverages cost-aware density to progressively divert selection away from saturated neurons and toward complementary parameter regions. Consequently, each new concept is assigned a sufficient yet non-overlapping subset, yielding smoother performance decay and improved stability over longer concept streams.

## 6. Conclusion and Future Work

We investigate continual personalization for diffusion models and identify a fundamental limitation of existing magnitude-based neuron selection methods: the absence of principled diversity constraints leads to semantic redun-dancy, parameter collisions, and amplified forgetting when learning semantically related concepts. To address this issue, we propose SCNS, which formulates cross-attention column adaptation as a constrained submodular maximization problem. By explicitly modeling coverage under diminishing returns via a Facility Location objective and incorporating cost-aware greedy selection, SCNS enables compact yet expressive updates with theoretical grounding in discrete optimization. Extensive experiments demonstrate that SCNS consistently outperforms state-of-the-art baselines in retention and identity fidelity while updating only a small fraction of parameters (0.41% per concept on average) and requiring no model fusion.

Despite its effectiveness, SCNS focuses on continual acquisition and retention rather than inference-time multi-concept composition, which may require additional binding and layout-control mechanisms. For very long concept streams, the available low-risk parameter subspace of a fixed backbone may also become saturated, causing later concepts to receive insufficient capacity. Future work may combine SCNS with composition-aware inference modules, adaptive capacity expansion, and multimodal knowledge editing.

## Acknowledgment

This work is supported by National Key R&D Projects (NO. 2024YFC3307100), NSFC Grant (No. 62576364), GuangDong Basic and Applied Basic Research Foundation (2026B1515020071), Shenzhen Basic Research Project (Natural Science Foundation) Basic Research Key Project (NO. JCYJ20241202124430041), the Open Research Fund from Guangdong Laboratory of Artificial Intelligence and Digital Economy (SZ) (NO. GML-KF-24-23), Shenzhen Science and Technology Program (NO. SYSRD20250529113401002), Xiaomi Technology Corporation, and the China Postdoctoral Science Foundation under Grant Number 2025M781535.

## Impact Statement

This work advances personalized generative AI by introducing a parameter-efficient, fusion-free framework for continual learning. A primary societal benefit of our approach is the enhancement of data privacy, by eliminating the need for data replay or the storage of historical user images, our method inherently reduces data leakage risks and facilitates compliance with privacy regulations. Furthermore, the extreme sparsity of our parameter updates contributes to democratizing AI and promoting "Green AI" by significantly lowering the energy costs and hardware barriers for model adaptation. However, we acknowledge that high-fidelity personalization carries potential risks of misuse, such as generating non-consensual imagery or unauthorized style mimicry. Additionally, as our method freezes the majority of the backbone, the model may retain intrinsic societal biases present in the pre-training data; thus, real-world deployment should be accompanied by robust content safety measures and ethical usage guidelines.

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

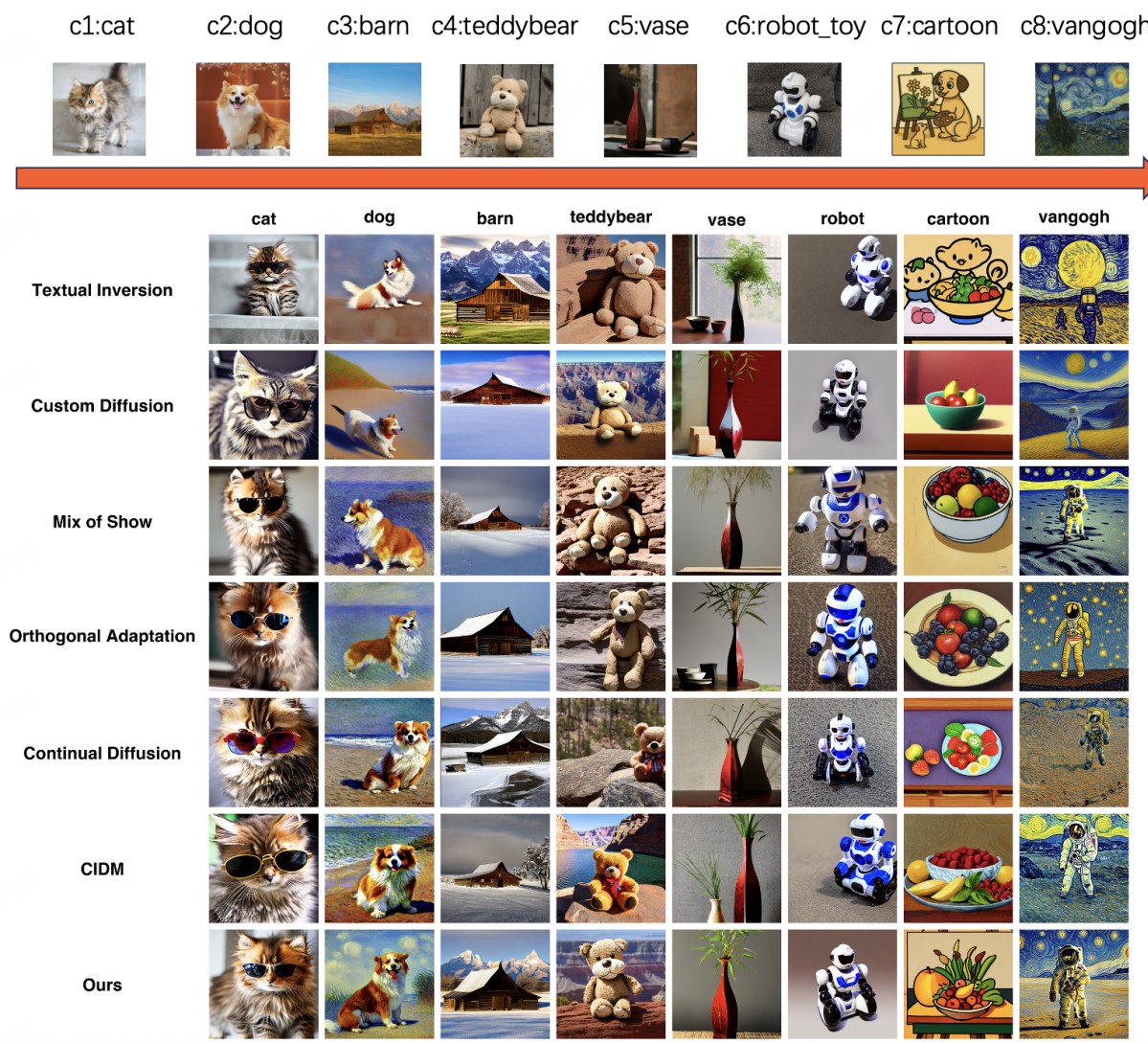

*Figure 6.* Qualitative comparison of continual personalization performance on an 8-concept stream. We evaluate the final model obtained after sequentially learning all concepts. For a rigorous comparison, images for each concept are generated using the *same specific prompt across all methods*, selected from our diverse evaluation set to cover various contexts (see details in **Appendix C.2**).

## A. Additional Supporting Materials

**qualitative comparisons**   This appendix provides additional visualizations that support the empirical findings in Sections 5.2. Specifically, Fig. 6 complements the qualitative comparisons by illustrating the final personalization quality after sequential learning.

**Stage-wise Concept-level Results**   To complement the averaged results in the main paper, we report the stage-wise concept-level performance of SCNS in Table 4. Here, "Stage" denotes the latest concept learned by the model checkpoint, and "Eval." denotes the concept being evaluated. These results provide a fine-grained view of how previously learned concepts are retained as new concepts are sequentially introduced.

*Table 4.* Stage-wise concept-level evaluation of SCNS. "Stage" denotes the latest concept learned by the model checkpoint, and "Eval." denotes the concept being tested.

| Stage | Eval. | CLIP-T | CLIP-I | DINO-I | Stage | Eval. | CLIP-T | CLIP-I | DINO-I |
|---|---|---|---|---|---|---|---|---|---|
| cat | cat | 0.2826 | 0.8574 | 0.7140 | robot | teddybear | 0.3213 | 0.8128 | 0.5882 |
| dog | cat | 0.2934 | 0.8606 | 0.7173 | robot | vase | 0.3102 | 0.7944 | 0.6059 |
| dog | dog | 0.2855 | 0.8320 | 0.7075 | robot | robot | 0.3069 | 0.7910 | 0.6065 |
| barn | cat | 0.2851 | 0.8567 | 0.7131 | Cartoon | cat | 0.2878 | 0.8610 | 0.7227 |
| barn | dog | 0.2972 | 0.8290 | 0.7128 | Cartoon | dog | 0.2931 | 0.8340 | 0.7049 |
| barn | barn | 0.2963 | 0.7745 | 0.6515 | Cartoon | barn | 0.2948 | 0.7750 | 0.6367 |
| teddybear | cat | 0.2810 | 0.8635 | 0.7067 | Cartoon | teddybear | 0.3199 | 0.8157 | 0.5920 |
| teddybear | dog | 0.2913 | 0.8360 | 0.7119 | Cartoon | vase | 0.3095 | 0.7890 | 0.6095 |
| teddybear | barn | 0.2966 | 0.7700 | 0.6561 | Cartoon | robot | 0.3036 | 0.7940 | 0.6098 |
| teddybear | teddybear | 0.3250 | 0.8122 | 0.5795 | Cartoon | Cartoon | 0.2887 | 0.7116 | 0.5854 |
| vase | cat | 0.2809 | 0.8635 | 0.7159 | Vangogh | cat | 0.2890 | 0.8570 | 0.7163 |
| vase | dog | 0.2995 | 0.8314 | 0.7020 | Vangogh | dog | 0.2935 | 0.8305 | 0.7034 |
| vase | barn | 0.2986 | 0.7670 | 0.6478 | Vangogh | barn | 0.2996 | 0.7730 | 0.6474 |
| vase | teddybear | 0.3245 | 0.8154 | 0.5831 | Vangogh | teddybear | 0.3191 | 0.8130 | 0.5809 |
| vase | vase | 0.3142 | 0.7905 | 0.6055 | Vangogh | vase | 0.3110 | 0.7895 | 0.6023 |
| robot | cat | 0.2809 | 0.8606 | 0.7139 | Vangogh | robot | 0.3016 | 0.7914 | 0.6090 |
| robot | dog | 0.2977 | 0.8260 | 0.7057 | Vangogh | Cartoon | 0.3014 | 0.7070 | 0.5792 |
| robot | barn | 0.2973 | 0.7653 | 0.6354 | Vangogh | Vangogh | 0.3187 | 0.7280 | 0.5565 |

# B. Additional Theory Analysis

## B.1. Additional Analysis of Fisher-weighted Preservation

This appendix details the theoretical construction of the Fisher-weighted preservation mechanism, focusing on the Fisher proxy estimation and the derivation of the unified risk function.

**Fisher Proxy as Hessian Approximation.** We estimate the importance of each cross-attention column $u$ at stage $p$ using a column-wise Fisher proxy:

$$f_p(u) \approx \mathbb{E}\big[g_p(u)^2\big].$$

This quantity approximates the diagonal of the Fisher Information Matrix (FIM), representing the expected curvature of the negative log-likelihood. From a Bayesian perspective, $f_p(u)$ corresponds to the precision of the posterior distribution; thus, units with high Fisher values encode historically certain knowledge, where perturbations are likely to incur significant loss increases (i.e., catastrophic forgetting).

**Interference Estimation via Gradient Proxy.** To assess gradient alignment without storing historical data, we approximate the inner product $\langle g_{\text{new}}, g_p \rangle$ by leveraging the Fisher proxy. Since $f_p(u)$ represents the expected squared gradient magnitude, we derive the first-order interaction as:

$$\text{dot\_proxy}_p(u) = \underbrace{\sqrt{f_p(u)}}_{\approx \|g_p\|} \cdot \|g_{\text{new}}(u)\|_2 \cdot \cos_p(u),$$

where $\cos_p(u)$ estimates the directional similarity. This allows SCNS to infer whether an update is distinctively constructive (positive cosine) or destructive (negative cosine) solely based on current gradients and the cached Fisher diagonal.

**Unified Risk Aggregation.** The final risk score integrates both directional interference (first-order) and curvature constraints (second-order) through a hinge-loss formulation:

$$\text{risk}_p(u) = \max\Big(0, \underbrace{-\text{dot\_proxy}_p(u)}_{\text{Directional Penalty}} + \alpha \underbrace{f_p(u)\,\|g_{\text{new}}(u)\|_2^2}_{\text{Curvature Penalty}}\Big).$$

The second term acts as a local Taylor expansion of the historical loss, ensuring that large updates to sensitive units are penalized even if directionally aligned. To handle multiple historical tasks, we aggregate these constraints using a smooth LogSumExp operator:

$$\text{risk}(u) = \frac{1}{\kappa} \log \sum_{p<i} \exp\big(\kappa \cdot \text{risk}_p(u)\big).$$

This differentiable soft-maximum emphasizes the most restrictive historical constraint, providing a robust safety margin for fusion-free personalization.

## B.2. Theoretical Analysis of Cost-aware Optimization

While the main text introduces the cost-aware module as a regularization term, this section provides a deeper theoretical analysis, framing it as a multi-objective optimization mechanism that balances immediate plasticity against long-term system sustainability.

**Mathematical Formulation of Gain Density.** Standard greedy algorithms for submodular maximization typically select the element $u^*$ that maximizes the marginal gain $\Delta F(u|S)$. However, in a continual learning setting, this approach is myopic as it ignores the *resource consumption* of parameters. SCNS redefines the selection criterion as a Gain Density optimization problem.

We define the cost of selecting a neuron $u$ as:

$$\text{Cost}(u) = c_0 + \alpha_{\text{ov}} \cdot \mathcal{H}(u),$$

where $c_0 = 1.0$ is the base selection cost, and $\mathcal{H}(u) \in [0, 1]$ represents the **historical overlap ratio** (the frequency with which unit $u$ has been used in previous tasks). Accordingly, the selection criterion shifts from maximizing gain to maximizing efficiency:

$$u^* = \operatorname*{argmax}_{u \in \mathcal{U} \setminus S} \frac{\Delta F(u|S)}{\text{Cost}(u)}.$$

The iterative process terminates when the maximum density falls below a threshold $\tau$, effectively identifying the stop point of the utility curve.

**Pareto Optimization Perspective.** This formulation solves a complex trade-off between three conflicting objectives, effectively finding an optimal solution on the Pareto Frontier defined by: (1) *New Task Plasticity* $\mathcal{O}_{\text{new}}$, (2) *Old Task Stability* $\mathcal{O}_{\text{old}}$, and (3) *Capacity Sustainability* $\mathcal{O}_{\text{future}}$.

- **Spatial Regularization via $\alpha_{\text{ov}}$:** The overlap penalty $\alpha_{\text{ov}}$ regulates the trade-off between $\mathcal{O}_{\text{new}}$ and $\mathcal{O}_{\text{old}}$. A high $\alpha_{\text{ov}}$ imposes a heavy tax on reused neurons. Even if a historical neuron offers a high gradient gain, its density may be lower than a fresh neuron with a moderate gradient. This forces the model to explore the unused parameter space, thereby prioritizing stability and reducing interference.

- **Efficiency Truncation via $\tau$:** The density threshold $\tau$ regulates the trade-off between $\mathcal{O}_{\text{new}}$ and $\mathcal{O}_{\text{future}}$. Due to the diminishing returns property of the submodular term, the marginal gain naturally decays. By stopping early at $\tau$, SCNS ensures the selected subset is "compact yet empirically effective". This prevents the model from wasting capacity on redundant features, effectively reserving parameters for $\mathcal{O}_{\text{future}}$.

**Dynamic Capacity Reservation.** The interaction between the cost function and the sequential stream creates an implicit dynamic capacity reservation mechanism. In the early stages, most neurons have $\mathcal{H}(u) \approx 0$, allowing the model to select the most salient neurons freely. However, as the sequence lengthens, high-gradient neurons accumulate overlap history, increasing their cost. Consequently, for later tasks, the algorithm is mathematically forced to bypass these saturated units and allocate parameters from the "long tail" of the distribution.

**Theoretical Analysis: Monotonicity and Approximation.** We strictly formulate the objective $F(S)$ as a monotone non-decreasing submodular function by transforming all regularization penalties into positive utility scores. Regarding optimization guarantees, while the density-based greedy strategy is known to approximate the optimal solution under knapsack constraints (bounded by $1 - 1/\sqrt{e}$ without partial enumeration (Lin & Bilmes, 2010)), our proposed threshold-based stopping rule ($\rho < \tau$) operates as a **Lagrangian relaxation** of the fixed-budget problem. This formulation ensures that the selected subset lies on the Pareto frontier of the utility-cost trade-off, guaranteeing that every updated parameter contributes a marginal efficiency strictly exceeding $\tau$, thereby adapting model capacity to semantic complexity.

# C. Implementation Details

## C.1. Datasets and Evaluation Protocol

To ensure a rigorous and fair comparison against state-of-the-art baselines, we constructed a standardized sequential image stream consisting of eight distinct concepts. The dataset stream is designed to cover a broad semantic span, necessitating the model to adapt to diverse domain shifts ranging from general objects to abstract artistic styles.

**Sequential Concept Stream.** The sequence is ordered as follows: `cat` $\rightarrow$ `dog` $\rightarrow$ `barn` $\rightarrow$ `teddybear` $\rightarrow$ `vase` $\rightarrow$ `robot_toy` $\rightarrow$ `Flat_Cartoon_Style` $\rightarrow$ `Van_Gogh_Style`. This selection includes dynamic animals, static objects, outdoor scenes, and global artistic styles, challenging the method's ability to maintain plasticity across significantly different semantic categories.

**Data Sources and Few-Shot Setting.** All training images are sourced from benchmark datasets established in pioneering personalization works, specifically Custom Diffusion[1] and DreamBooth[2]. These datasets serve as the standard foundation for recent continual personalization research. Consistent with the few-shot personalization setting, we utilize only **5 to 6 reference images** per concept. This strictly constrained data regime tests the data efficiency of the algorithm and its ability to extract identities without overfitting.

**Evaluation Prompts.** To comprehensively evaluate generation quality and alignment, we curated a fixed set of **20 diverse text prompts** for each concept (totaling 160 prompts for the full stream). These prompts range from simple re-contextualization to complex compositional tasks, ensuring that evaluation metrics reflect robust generalization rather than memorization. We emphasize that all experiments—including SCNS and all baselines—were trained and evaluated strictly under this identical data stream and ordering.

**Sampling and Scoring Protocol.** To ensure robust quantitative results, we employed two distinct sampling protocols for evaluation:

- **Final Quality Assessment:** After training the complete stream of 8 concepts, we generated 10 images for each of the 20 evaluation prompts per concept. This results in a total test set of 1,600 images (200 images $\times$ 8 concepts) used to compute the final CLIP and DINO scores reported in Tab. 1.

- **Continual Forgetting Analysis:** To track performance degradation, we performed a cumulative evaluation at the end of each learning stage. At stage $t$, we generated samples for the current concept $c_t$ and all prior concepts $\{c_1, \ldots, c_{t-1}\}$. Across the full 8-stage stream, this constitutes a comprehensive evaluation of 7,200 images ($200 \times \sum_{t=1}^{8} 1 \approx 200 \times 36$), ensuring a rigorous measurement of the Task Forgetting Rates (TFR).

## C.2. Prompts for Visualization (Fig. 6)

In the main qualitative comparison (Fig. 6, we selected one representative prompt for each concept from our evaluation set to demonstrate the model's capability across different generation tasks. The specific prompts used are:

- **Concept 1 (cat):** ``<new1> cat wearing sunglasses''

- **Concept 2 (dog):** ``An art painting of <new2> dog on the beach''

---

[1] https://github.com/adobe-research/custom-diffusion
[2] https://github.com/google/dreambooth

- **Concept 3 (barn):** ``<new3> barn in snowy ice''

- **Concept 4 (teddybear):** ``<new4> teddybear in Grand Canyon''

- **Concept 5 (vase):** ``Bamboo in <new5> vase''

- **Concept 6 (robot_toy):** ``Photo of <new6> robot_toy''

- **Concept 7 (Flat_Cartoon_Style):** ``A plate of fruits in <new7> style''

- **Concept 8 (Van_Gogh_Style):** ``an astronaut on the moon in <new8> style''

## C.3. Analysis of Universal Semantic Anchors

To calculate the Task Relevance score $s_{\text{rel}}(u)$ (Eq. 7), we employ a set of Universal Semantic Anchors ($\mathcal{A}$) to estimate the general responsiveness of neurons. Here, we provide details on the construction of $\mathcal{A}$, its sensitivity, and an ablation study justifying the Signal-to-Noise Ratio (SNR) formulation.

**Construction and Composition.** The anchor set consists of $|\mathcal{A}| = 100$ distinct text prompts designed to cover the general semantic manifold of the pre-trained Stable Diffusion model. Unlike random text generation, we manually curated these prompts to ensure they trigger valid, high-probability activations across four major semantic categories: **Common Objects**, **Artistic Styles**, **Scene/Landscapes**, and **Actions/Compositions**.

The composition is detailed in Tab. 5. This diversity ensures that the denominator in our relevance score, $\|g_{\text{gen}}(u)\|_2$, accurately reflects a neuron's "general" sensitivity to broad concepts, allowing $s_{\text{rel}}$ to isolate "specialist" neurons (high response to specific concept, low response to general anchors) from "generalist" hubs.

*Table 5.* Composition of the Universal Semantic Anchors (USA) set ($|\mathcal{A}| = 100$).

| Category | Count | Representative Examples |
|---|---|---|
| Common Objects | 30 | *"A photo of a dog", "A photo of a laptop", "A photo of a tree"* |
| Artistic Styles | 20 | *"An oil painting", "A cyberpunk illustration", "A sketch drawing"* |
| Scenes | 25 | *"A forest landscape", "A modern art gallery", "A starry night sky"* |
| Actions | 25 | *"A person walking", "Waves crashing", "A musician playing guitar"* |

## C.4. SCNS Hyperparameter Settings

The hyperparameters for our proposed Submodular Concept Neuron Selection (SCNS) were determined through extensive ablation studies. We detail both the balancing coefficients and the specific settings for the greedy selection strategy below:

**Objective Balancing Coefficients.**

- **Task Relevance ($\lambda_{rel} = 1.0$):** Controls the importance of the gradient norm for the new concept.

- **Preservation Risk ($\lambda_{pres} = 0.5$):** Penalizes updates to parameters with high Fisher information to protect prior knowledge.

- **Submodular Span ($\lambda_{span} = 1.0$):** Encourages selection of neurons providing comprehensive semantic coverage.

- **Zero-shot Protection ($\lambda_{zero} = 0.3$):** Regularizes the magnitude of weight changes.

**Selection Strategy Parameters.**

- **Semantic Dimensions ($K = 64$):** For the Facility Location term (Eq. 9), we utilize the top-64 dimensions of the CLIP text embeddings. This value was empirically chosen to cover the majority of salient semantic features without introducing excessive noise.

- **Cost-Aware Parameters** ($c_0 = 1.0, \alpha_{\text{ov}} = 2.0$): We set a base cost $c_0 = 1.0$ to enforce a fundamental utility threshold for selection, combined with an overlap penalty $\alpha_{\text{ov}} = 2.0$ to actively discourage historical neuron reuse. This joint configuration effectively steers the model towards fresh parameter subspaces while ensuring that every selected unit provides sufficient marginal gain to justify its occupation.

- **Density Threshold** ($\tau = 0.2$): The adaptive stopping threshold is set to $\tau = 0.2$. This value ensures the algorithm terminates when the marginal gain density becomes insufficient, preventing over-allocation of parameters.

**C.5. Baseline Configurations**

To ensure fair comparisons, we strictly adhered to the hyperparameter settings and protocols described in the original papers or official codebases. The baselines are categorized into two paradigms: *Train-then-Merge* and *Sequential Continual Learning*.

C.5.1. MERGING-BASED APPROACHES

These methods optimize each concept independently and subsequently consolidate them into a unified model.

**Textual Inversion (TI).** We utilized the official code repository[3]. For each concept, we optimized a new specific token embedding within the tokenizer. The learning rate was set to $5 \times 10^{-4}$, and training was fixed at 1,500 steps. After training all concepts individually, we merged the learned embeddings into a single embedding file for inference.

**Custom Diffusion (CD).** Following the official implementation[4], we fine-tuned both the new text embeddings and the key/value ($W_k, W_v$) projection layers. The learning rate was set to $1 \times 10^{-5}$. We trained each concept for 500 steps and saved only the delta weights ($\Delta W$). Finally, we employed the official optimization-based merging script to consolidate the delta weights of all eight concepts. Additionally, we report sequential-CD (no merge) in Tab. 2, trained by fine-tuning text embeddings and $W_k, W_v$ sequentially across concepts without applying the merging script.

**Mix-of-Show.** We adopted the official implementation[5]. A task-specific LoRA module was trained for each concept. The learning rates were set to $1 \times 10^{-3}$ for text embeddings and $1 \times 10^{-4}$ for LoRA weights. Consistent with the official setting, training steps were dynamically determined based on the number of images. We used the provided gradient fusion script to combine the LoRA blocks; we note that this fusion process is computationally intensive, taking approximately 40–50 minutes on a standard GPU.

**Orthogonal Adaptation.** As the official code is unavailable, we re-implemented this method following the paper's details. We utilized the recommended randomized orthogonal basis constraints. The learning rates were set to $1 \times 10^{-3}$ for text embeddings and $1 \times 10^{-5}$ for LoRA layers, with training fixed at 500 steps per concept.

C.5.2. SEQUENTIAL CONTINUAL LEARNING APPROACHES

These methods learn concepts sequentially, where the model at stage $i$ is initialized from the weights of stage $i - 1$.

**Continual Diffusion (C-LoRA).** We re-implemented the method, incorporating the *self-regularization loss* to mitigate forgetting. We set the text embedding learning rate to $1 \times 10^{-3}$ and the LoRA learning rate to $1 \times 10^{-5}$. Each concept was trained for 500 steps sequentially. Unlike merging-based methods, the training for the $n$-th concept directly fine-tunes the checkpoint from the $(n - 1)$-th concept.

**CIDM.** We utilized the official GitHub repository[6] and followed its default configuration. Similar to Continual Diffusion, CIDM adopts a sequential incremental learning protocol, using the default learning rates and training steps specified in the codebase.

---

[3] https://github.com/rinongal/textual_inversion
[4] https://github.com/adobe-research/custom-diffusion
[5] https://github.com/TencentARC/Mix-of-Show
[6] https://github.com/jiahuadong/cifc

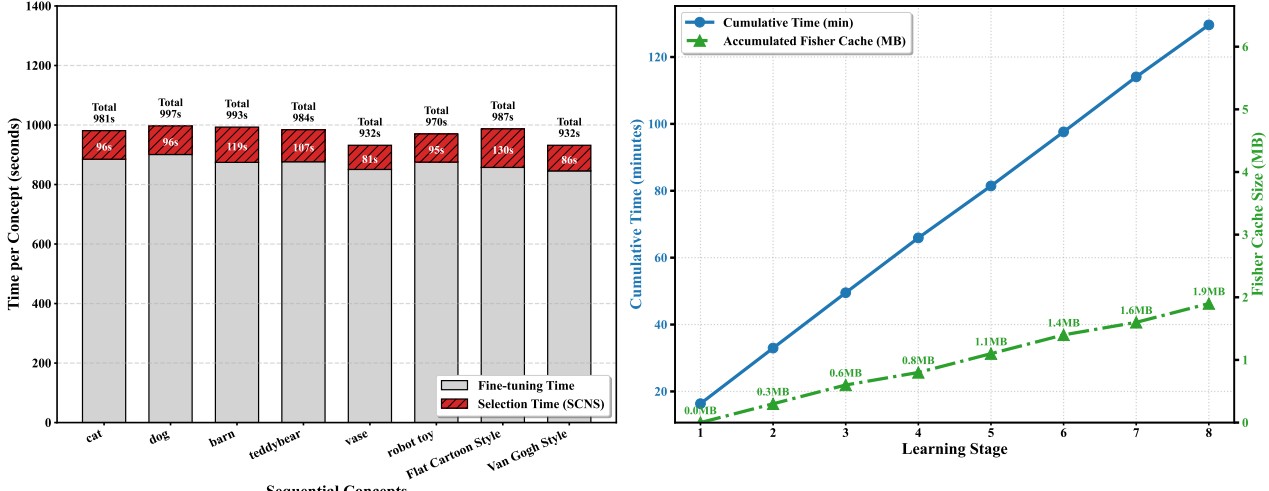

*Figure 7.* **Computational Efficiency and Scalability Analysis. Left:** Per-concept runtime breakdown. The Selection Time (SCNS) is explicitly annotated inside the bars, averaging roughly **1.5 to 2 minutes**, which constitutes a minor fraction ($\sim$10%) of the total fine-tuning budget. **Right:** Cumulative resource consumption. The Fisher Cache storage grows linearly but remains negligible, occupying only **1.9 MB** after learning the full 8-concept stream. This demonstrates that SCNS is highly efficient in both time and storage.

**Concept Neuron Selection (CNS).** As the official code is not publicly available, we attempted to re-implement the method strictly following the algorithmic details provided in the original paper. Our implementation identifies the adaptation subset by selecting neurons that exhibit high gradient magnitudes for the target concept, while explicitly filtering out those showing high responsiveness to a set of generic prompts (to isolate concept-specific neurons). The selected subset is then fine-tuned incorporating the specific regularization loss defined in CNS. Consistent with our experimental protocol, we set the learning rate to $5 \times 10^{-4}$ for text embeddings and $3 \times 10^{-5}$ for concept neurons, with the training duration fixed at 500 steps per concept.

### C.6. Computational Complexity and Overhead

We provide a detailed quantification of the computational, storage, and memory overhead introduced by SCNS, based on experiments conducted on a server equipped with two NVIDIA RTX A6000 GPUs (48GB each). The efficiency analysis is summarized in Fig. 7.

- **Storage Efficiency (Fisher Cache):** SCNS maintains a cache for the diagonal of the Fisher Information Matrix (FIM) and EMA gradients for the selected sparse subset. As shown in Fig. 7 (Right), this cache is extremely compact. For the 8-concept stream, the cumulative storage occupies only **1.9 MB**. Even if scaled to dozens of concepts, the storage footprint remains negligible compared to replay-based methods that require storing gigabytes of latent features or generated images.

- **Runtime Efficiency:** The primary computational cost lies in the iterative submodular selection process. According to our logs (Fig. 7 Left), the neuron selection phase takes approximately **80 to 130 seconds** (approx. 1.5–2 minutes) per concept. Once the subset is selected, fine-tuning follows the standard efficient protocol ($\sim$15 minutes for 500 steps). The total turnaround time per concept is significantly faster than post-hoc fusion methods like Mix-of-Show, which often require 40–50 minutes of optimization-based merging.

- **GPU Memory Consumption:** During the training phase, the peak Video RAM (VRAM) usage is approximately **34 GB per GPU** on our dual-A6000 setup (using a batch size of 2 per GPU with gradient accumulation). Since SCNS freezes the vast majority of the backbone parameters and only updates a sparse subset ($< 0.5\%$ of parameters), the memory overhead for optimizer states is minimized, making it feasible for standard high-end workstations.

# D. Additional Experiments Analyses

## D.1. Sufficiency via Diminishing Returns and Cost Regularization

The divergence in performance on the extended 10-concept stream highlights a critical interaction between the submodular objective and the cost constraint. In the ablation setting Fig. 5, the reliance solely on magnitude-based gains leads to a *redundancy trap*: the algorithm repeatedly selects "hub neurons" that offer high initial gradients but rapidly diminishing semantic returns. Once these neurons are saturated by earlier tasks, the selection for new concepts (`teapot`, `backpack`) becomes insufficient—it fails to capture the unique fine-grained features required, leading to the observed overwrite of style concepts.

In contrast, the full SCNS leverages the *diminishing returns property* to guarantee **subset sufficiency**. By incorporating the cost term, we modify the selection density to Eq. 11. As the most active neurons accumulate overlap cost, their density decreases sharply. Crucially, this mechanism forces the greedy search to escape the local optima of high-magnitude neurons and explore the *semantically complementary* parameter space. Even if alternative neurons have lower absolute gradients, their **marginal gain** relative to the uncosted budget remains high. This ensures that the selected subset is not just a repetition of dominant features, but a sufficiently diverse collection that covers the full semantic span of the new concept without encroaching on the territory of previous tasks.

## D.2. Curriculum Order Robustness Analysis

A common failure mode in continual learning is sensitivity to the curriculum order (De Lange et al., 2021). Models often perform significantly differently depending on whether "difficult" concepts are learned early or late in the sequence. To evaluate the robustness of SCNS against concept permutation, we evaluated the model on three distinct random orderings of the 8-concept stream, as detailed in Tab. 6.

**Experimental Results.** The quantitative results demonstrate high stability across all permutations.

- **Minimal Variance:** The standard deviation across the three runs is negligible: $\sigma \approx 0.0031$ for CLIP-Text, $\sigma \approx 0.0011$ for CLIP-Image, and $\sigma \approx 0.0055$ for DINO-Image.

- **Consistent Fidelity:** The Image Alignment (CLIP-I) remains consistently high ($\sim 0.7852$) regardless of whether complex styles (e.g., `Van_Gogh`) appear at the end (Sequence C) or in the middle (Sequence B).

- **Identity Preservation:** DINO-I scores show slight fluctuations (ranging from $0.6216$ to $0.6348$), indicating that identity preservation remains robust even when the concept is introduced at different stages of network saturation.

*Table 6.* Robustness analysis across different curriculum orders. The model achieves consistent performance metrics (Mean CLIP-T/I and DINO-I across all 8 concepts) regardless of the learning sequence, validating the efficacy of our adaptive capacity allocation.

| ID | Sequence Order | CLIP-T ↑ | CLIP-I ↑ | DINO-I ↑ |
|---|---|---|---|---|
| **Seq A** | Barn → Teddy → Robot → Vase → Cat → Dog → VanGogh → Cartoon | 0.3012 | **0.7864** | **0.6348** |
| **Seq B** | Teddy → Cat → Dog → Robot → VanGogh → Vase → Barn → Cartoon | 0.2966 | 0.7838 | 0.6300 |
| **Seq C** | Cat → Dog → Barn → Teddy → Vase → Robot → Cartoon → VanGogh | **0.3041** | 0.7854 | 0.6216 |
| *Std. Dev.* | - | *0.0031* | *0.0011* | *0.0055* |

**Reason for Robustness.** We attribute this order-agnostic behavior to two key mechanisms in SCNS:

1. **Adaptive Stopping ($\tau$):** Unlike fixed-budget methods that strictly limit parameter usage per step, our density-based stopping criterion dynamically allocates more neurons if a concept is semantically complex or "hard", regardless of its position in the sequence.

2. **Cost-Aware Overlap Penalty ($\alpha_{ov}$):** The overlap penalty ensures that later tasks (which face a more saturated parameter space) are actively steered towards fresh, unused neurons. This prevents the "concept crowding" effect often seen in sequential learning, ensuring that late-arriving concepts do not destructively overwrite early ones.

## D.3. Cross-Architecture Extension: Application to Stable Diffusion 3

To demonstrate the versatility and future-proof nature of SCNS, we extended our framework from the U-Net-based Stable Diffusion v1.5 (SD1.5) to Stable Diffusion 3 (SD3) (Esser et al., 2024), which employs a Multimodal Diffusion Transformer (MM-DiT) architecture. This section analyzes the architectural shifts, adaptation details, and empirical results.

### D.3.1. ARCHITECTURAL SHIFTS: FROM U-NET TO MM-DIT

The transition from SD1.5 to SD3 involves fundamental changes in backbone topology and conditioning mechanisms, as summarized in Tab. 7.

**Backbone Structure.** SD1.5 relies on a convolutional U-Net where cross-attention layers are interspersed between ResNet blocks. In contrast, SD3 utilizes a pure Transformer architecture (MM-DiT). Text and image modalities are tokenized into separate sequences but interact via joint attention blocks. Despite this topological shift, the mechanism for injecting semantic conditions remains mathematically consistent: both architectures employ learnable projection matrices $(W_k, W_v)$ to map text embeddings into the visual latent space.

**Semantic Space Dimensionality.** A critical difference lies in the text encoding. SD1.5 uses a single CLIP encoder ($D = 768$). SD3 integrates three encoders (CLIP-L, CLIP-G, and T5-XXL (Raffel et al., 2023), expanding the semantic space dimension to $D = 4096$. This 5x increase in dimensionality implies a significantly sparser activation pattern for any single concept, theoretically making our submodular selection even more effective.

*Table 7.* Key architectural differences between SD1.5 and SD3 and their implications for SCNS.

| Feature | Stable Diffusion v1.5 | Stable Diffusion 3 (SD3) |
|---|---|---|
| **Backbone** | Convolutional U-Net | Multimodal Diffusion Transformer (MM-DiT) |
| **Attention Target** | Cross-Attention (`to_k`, `to_v`) | Joint Attention (`add_k_proj`, `add_v_proj`) |
| **Text Dimension** | 768 (CLIP ViT-L/14) | 4096 (T5-XXL + CLIPs) |
| **Training Objective** | Noise Prediction (DDPM) | Flow Matching (Rectified Flow) |
| **SCNS Adaptation** | Selects from ~16 layers | Selects from 38 Joint-Attn blocks |

### D.3.2. ADAPTATION OF SCNS TO MM-DIT

We adapted the SCNS algorithm to the SD3 architecture with the following modifications:

1. **Target Identification:** Instead of searching for `CrossAttention` modules, SCNS targets the `JointAttention` blocks in the MM-DiT. Specifically, we apply selection to the `add_k_proj` and `add_v_proj` layers, which process the 4096-dimensional text conditions.

2. **High-Dimensional Submodularity:** The semantic universe $\mathcal{D}$ in the Facility Location term (Eq. 9) is expanded to cover the top-$K$ dimensions of the T5-XXL embedding. Given the higher dimensionality ($D = 4096$), we empirically found that increasing $K$ to 128 ensures robust coverage of the richer semantic features.

3. **Flow Matching Objective:** The gradient signatures $g_u$ are computed using the Flow Matching loss ($\mathcal{L}_{\text{flow}} = \|v_\theta(z_t) - (x_1 - x_0)\|^2$) rather than the standard noise prediction loss (Lipman et al., 2022).

### D.3.3. EXPERIMENTAL RESULTS AND ANALYSIS

We replicated the 8-concept continual learning benchmark on SD3 using the adapted SCNS method. **Notably, we directly applied the hyperparameter settings and selection protocols optimized for SD1.5 to SD3 without specific tuning**, demonstrating the method's robust transferability across architectures.

**Quantitative Performance.** SCNS achieves a CLIP-T score of 0.3288, representing a consistent improvement of 0.01–0.02 over the SD1.5 implementation. This indicates that the MM-DiT backbone significantly enhances prompt adherence and text-image alignment. However, we observe a slight decrease in identity preservation metrics, with CLIP-I at 0.7746 and DINO-I at 0.6096 (compared to $\sim 0.7854$ and $\sim 0.6216$ on SD1.5). We attribute this to the semantic inertia of the stronger

foundation model: while SD3's powerful generative prior enables richer background synthesis and more intricate details, its massive parameter space is inherently more resistant to drastic distribution shifts towards the few-shot reference data than the more malleable SD1.5 U-Net.

**Qualitative Results.** As shown in Fig. 8, the generated images exhibit exceptional fidelity and detail. Benefiting from SD3's superior backbone, SCNS renders complex prompts (e.g., context changes) with greater accuracy than SD1.5. For instance, the Van_Gogh_Style (Concept 8) displays nuanced brushwork textures while maintaining the structural integrity of the scene. Although the identity alignment scores are slightly lower, the visual results suggest a favorable trade-off where the model generates more diverse and aesthetically pleasing compositions rather than overfitting to the reference background.

**Why SCNS is Effective on DiT?** The success of SCNS on SD3 validates the "Universality of Attention" hypothesis. Although the network depth increases and the embedding dimension explodes (4096 vs 768), the principle of "sparse, column-wise adaptation" holds true. In fact, the sparsity ratio is even more favorable on SD3 due to the massive parameter count of the MM-DiT (2B+ params). This confirms that submodular neuron selection is a scalable strategy for efficiently personalizing large-scale foundation models.

**Note.** Existing continual learning baselines have not yet been applied to the latest Stable Diffusion 3 architecture in the literature. While we attempted to adapt these methods to the MM-DiT backbone, the preliminary results were suboptimal. Consequently, we omit these comparisons to focus on the zero-shot transferability of SCNS.

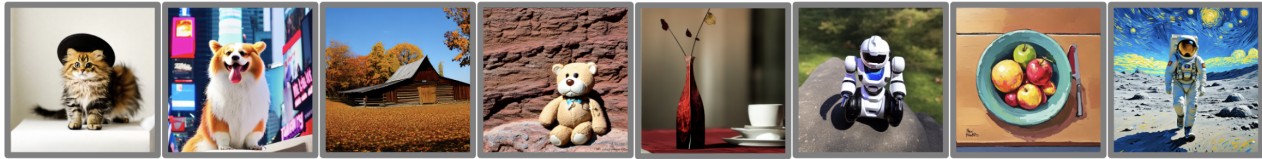

*Figure 8.* **Zero-shot Transfer to SD3 Architecture.** Qualitative results of SCNS applied to the 8-concept stream on the Stable Diffusion 3 (2B) backbone. Despite the "inertia" of the large model leading to slightly lower numeric identity scores, the method generates images with superior prompt adherence and rich details, demonstrating robust continual capabilities on Transformer-based diffusion models.

### D.4. Parameter Efficiency and Sparsity Quantifications

To strictly quantify the parameter efficiency of SCNS, we track the number of trainable parameters across a 10-concept sequential stream.

**Per-Concept Sparsity.** Tab. 8 details the exact parameter usage for each stage. On average, SCNS updates approximately **0.41%** of the total model parameters per concept (ranging from 0.81% for the initial concept to ∼0.22% for later concepts). This validates our claim of extreme sparsity compared to standard fine-tuning or LoRA-based approaches.

**Cumulative Footprint.** Fig. 10 visualizes the cumulative parameter growth. After learning 10 diverse concepts, the total unique parameter footprint is **21.03 M**, which constitutes only **2.45%** of the full U-Net capacity (859.52 M). The sub-linear growth of the cumulative curve indicates that SCNS effectively structured sharing existing selected neurons, ensuring the model remains compact even as the number of tasks increases.

## E. Limitations

Although SCNS effectively balances plasticity and stability through submodular optimization, several limitations remain. First, the scoring mechanism relies on heuristic proxies, specifically the SNR-based gradient metric for task relevance and the use of the top $K$ semantic dimensions for semantic coverage. Balancing these objectives involves hyperparameters (e.g., coefficients $\lambda$, density-based threshold $\tau$, and overlap penalty $\alpha_{ov}$) that currently require empirical tuning through exponential search, lacking a fully dynamic adaptive mechanism.

*Table 8.* Detailed sparsity quantification per concept. "Sparsity" denotes the percentage of trainable parameters relative to the full U-Net.

| Stage | Concept | Selected Columns | Trainable Params | Sparsity (%) |
|---|---|---|---|---|
| 1 | cat | 3,921 | 6.96 M | 0.810 |
| 2 | dog | 2,849 | 4.69 M | 0.546 |
| 3 | barn | 1,938 | 3.30 M | 0.384 |
| 4 | teddybear | 1,842 | 3.10 M | 0.361 |
| 5 | vase | 1,861 | 2.88 M | 0.336 |
| 6 | robot_toy | 1,098 | 1.86 M | 0.217 |
| 7 | Flat_Cartoon | 2,703 | 4.54 M | 0.529 |
| 8 | Van_Gogh | 2,093 | 3.33 M | 0.388 |
| 9 | teapot | 1,581 | 2.36 M | 0.275 |
| 10 | backpack | 1,337 | 1.90 M | 0.221 |
| **Avg.** | - | **2,122** | **3.49 M** | **0.41%** |

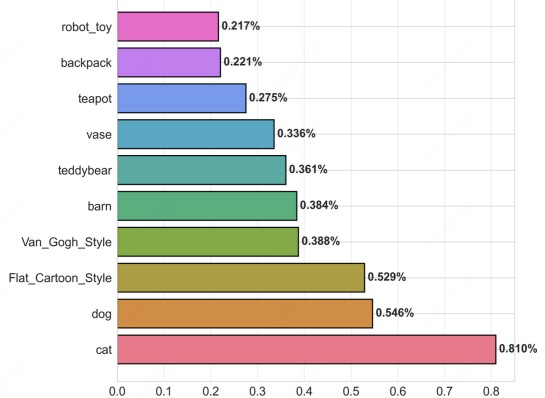

*Figure 9.* **Subset Size vs. Stage.** Number of parameters allocated for each new concept.

*Figure 10.* **Cumulative Footprint.** Total unique parameters updated across the stream.

Another major limitation of this study is the lack of dedicated optimization for multi-concept prompt composition. In such scenarios, the model may face attribute binding or concept loss issues. We differentiate continual acquisition from combinatorial composition: **Continual Acquisition Task**: The core of this study, aimed at ensuring high-quality learning of new concepts while preventing forgetting of old ones, with a focus on parameter space allocation and protection. **Combinatorial Composition Task**: Focuses on resolving semantic conflicts and layout control between different concepts during inference. Currently, many baseline methods address the composition problem by modifying attention mechanisms or using layout control. While SCNS can seamlessly integrate with these inference-side techniques, its core design focuses on parameter update logic (Selection-then-Finetuning), rather than spatial layout control. The combination of SCNS's neuron selection capability with existing region-based reasoning techniques will be an important direction for future continual diffusion models.

