# OpenReview forum: "SCNS: Continual Personalization of Diffusion Models via Submodular Concept Neuron Selection"
_ICML.cc/2026/Conference — ICML 2026 regular_

### Official Review · Reviewer_zLjG · 2026-02-15

**Soundness:** 3
**Presentation:** 3
**Significance:** 2
**Originality:** 2
**Overall Recommendation:** 4
**Confidence:** 1

**Summary:**

This paper proposes SCNS, a method designed to address the problem of continual personalization of diffusion models, enabling them to efficiently learn new concepts while avoiding forgetting old ones. SCNS formulates continual personalization as a constrained submodular optimization problem to select a minimal yet sufficient set of concept-specific neurons, taking into account the diminishing marginal returns of parameters. SCNS integrates a facility-location-based coverage objective to suppress semantic redundancy, employs a Fisher-weighted risk proxy to protect previously learned concepts, and adopts a cost-aware greedy rule to balance stability and plasticity while maintaining high sparsity.

**Compliance With Llm Reviewing Policy:**

Affirmed.

**Final Justification:**

I have declared myself a non-expert in this field and decided to maintain a "Weak Accept" score of 4 while keeping the confidence score at 1.

**Key Questions For Authors:**

See Weaknesses.

**Limitations:**

Yes.

**Strengths And Weaknesses:**

Strengths:

(1) Continual personalization of diffusion models is an important research direction.

(2) Applying submodule optimization to the continual personalization of diffusion models is an effective approach.

(3) Extensive experiments demonstrate that SCNS achieves superior performance in terms of image alignment and anti-forgetting.

Weaknesses:

I am completely unfamiliar with this field, so it is difficult for me to point out specific shortcomings. I only list some of my questions below.

(1) What are the differences between continual personalization of diffusion models and continual learning in general machine learning? What issues require extra attention?

(2) Which paradigm of continual learning does the methodological framework of continual personalization of diffusion models belong to?

---

> ### Author Rebuttal · Authors · 2026-03-30
>
> We thank the reviewer for the positive assessment of the paper’s technical quality, thorough empirical evaluation, and the importance of continual personalization for diffusion models. The two questions are also very insightful, and we agree that clarifying this connection will make the paper more accessible, especially to readers outside the continual diffusion personalization community.
>
> ### Q1. Difference between continual personalization of diffusion models and general continual learning.
>
> Continual personalization shares the core goal of general continual learning—learning new tasks without forgetting previous ones—but it also introduces several challenges that require extra attention.
>
> First, the objective is not standard classification or representation learning, but subject-/concept-driven generation: the model must preserve both text alignment and identity fidelity of previously learned personalized concepts under diverse prompts. Accordingly, its failure modes are also different from standard accuracy degradation; instead, they appear as identity drift, style drift, concept vanishing, attribute leakage, and cross-concept interference. This is exactly why we evaluate the method jointly with CLIP-T, CLIP-I, DINO-I, and TFR, rather than relying on a single forgetting metric.
>
> Second, the supervision is typically few-shot, with only 5–6 reference images per concept in our benchmark, which makes learning new concepts substantially more challenging. This is also why our method emphasizes parameter-efficient and selective adaptation: instead of broadly updating the model under extremely limited supervision, SCNS identifies a compact subset of concept-relevant neurons so that the available evidence can be used more effectively while reducing unnecessary interference.
>
> Third, continual personalization is tightly tied to the architecture of diffusion models. Because text-image alignment is primarily controlled by shared cross-attention layers, and multiple concepts are inserted sequentially into the same conditional denoising model, interference arises not only at the task level but also at the level of fine-grained semantic attributes and overlapping parameter subspaces. This naturally leads to semantic redundancy, parameter collisions, and destructive interference. This is also why our method focuses on sparse column-wise neuron selection over the key/value projections of cross-attention layers in the diffusion backbone.
>
> In short, continual personalization can be viewed as a specialized continual learning problem for generative foundation models, but with additional constraints from few-shot concept learning and identity preservation.
>
> ### Q2. Which continual learning paradigm does this work belong to?
>
> We agree that this should be stated more explicitly. Our setting can be described as replay-free and parameter-efficient sequential adaptation.
>
> From the perspective of the training paradigm, continual personalization is closest to replay-free task-incremental continual learning. Concepts arrive sequentially; each stage introduces a new personalized concept token; and the same shared model must preserve all previously learned concepts without revisiting historical data. This sequential concept stream is exactly the training protocol illustrated in Fig. 1 and used throughout our experiments. This is also why our setting differs from merge-based baselines: although their broader goal is also multi-concept adaptation, they train each concept independently as a separate model or adapter block and only combine them afterward, rather than learning in a strictly sequential manner.
>
> From the perspective of the learning mechanism, SCNS is most closely related to soft parameter isolation. More specifically, the submodular neuron selection mechanism allocates sparse and low-overlap neuron subsets to different concepts, encouraging each concept to occupy its own parameter subspace as much as possible and thereby reducing interference from later updates.
>
> We appreciate this suggestion. We would be happy to incorporate a clarified version of this discussion into the Related Work section as well, as it would make the paper easier to follow for readers from the broader continual learning community.

---

> > ### Author Rebuttal · Reviewer_zLjG · 2026-04-01
> >
> > Thank you for your response. Continual personalization of diffusion models is a novel and interesting field for me. To avoid influencing the judgment of other reviewers, I have declared myself a non-expert in this field and decided to maintain a "Weak Accept" score of 4 while keeping the confidence score at 1.

---

> > > ### Author Response · Authors · 2026-04-03
> > >
> > > Thank you for the thoughtful follow-up and for your careful and fair assessment. We appreciate your openness about your background in the area, and we are glad the rebuttal was helpful. We will incorporate these clarifications into the revised manuscript to make the paper more accessible to readers from the broader continual learning community.

---

### Official Review · Reviewer_1MAP · 2026-03-07

**Soundness:** 3
**Presentation:** 3
**Significance:** 3
**Originality:** 3
**Overall Recommendation:** 4
**Confidence:** 3

**Summary:**

This paper addresses the problem of personalization in diffusion models and proposes a neuron-level concept modification method. The authors design a series of metrics to guide the unit selection, including signal-to-noise ratio as a relevance score to the current concept, Fisher penalty as a preservation score to previous concepts, signal magnitude alignment to ensure the locality of the selection process,  and a safety score to encourage the selection of units with smaller magnitudes. By applying these metrics to guide a greedy selection process, the method identifies a subset of neurons for sparse updates. The proposed method demonstrates better performance in quantitative analysis.

**Compliance With Llm Reviewing Policy:**

Affirmed.

**Final Justification:**

I would strongly suggest avoiding the term “compact.” It still implies a restrictive condition and may lead to overclaim without sufficient justification.

**Key Questions For Authors:**

Please refer to the weaknesses.

**Limitations:**

Yes, the authors have adequately discussed the limitations and potential negative societal impact of their work

**Strengths And Weaknesses:**

### Strengths
- The paper is well written and easy to follow. Given an input concept, it first formulates model personalization as the problem of selecting an optimal subset of neurons, explicitly considering the joint effect of different neuron combinations on the target objective. By enforcing global optimality, this problem formulation well captures the correlations among neurons, avoiding the need for explicitly modeling or defining inter-neuron relationships.
- The fine-grained neuron-level update strategy is novel and consistent with intuition, where neurons show correlation and thus should not be evaluated in isolation. To this end, it focuses on neuron units for different input concepts, following a more reasonable and self-motivated setting that may inspire future research.
- The proposed method is simple and effective, where the guiding metrics show effectiveness in identifying units to achieve continual personalization while avoiding catastrophic forgetting.

### Weaknesses
- The paper introduces a series of metrics to guide the neuron selection process. The authors should provide ablation studies to analyze the contribution of each metric to the overall performance.
- In L92–93, the authors claim that the selected neuron subset is “minimal and sufficient.” How can this claim be rigorously proven?
- The evaluation results only report overall fine-tuning performance. How does the fine-grained tuning process affect personalization success rate, generalization to different expressions of the same concept, and potential interference with other concepts?

Overall, I think this paper has its merits despite its insufficiency regarding empirical evaluation. I will reassess my score after considering the authors’ response during the rebuttal phase.

---

> ### Author Rebuttal · Authors · 2026-03-30
>
> We thank the reviewer for the constructive feedback and for recognizing the motivation and effectiveness of our neuron-level continual personalization framework. We address the three main concerns below.
>
> ### W1. Ablation on each metric
>
> We agree that the role of each scoring component should be isolated more explicitly. We therefore conducted an additional component ablation, setting each weight to zero in turn. As shown below, removing any one term leads to a measurable performance drop on at least part of the evaluation, supporting the complementary roles of the four components.
>
> | |clip_text|clip_image|dino_image|TFR-clip|TFR-dino|
> |---|---|---|---|---|---|
> |$\lambda_{rel}=0$|0.3026|0.7684|0.5893|3.7251|5.3487|
> |$\lambda_{pres}=0$|0.2976|0.7774|0.6099|1.4342|2.4477|
> |$\lambda_{span}=0$|0.2924|0.7711|0.6057|0.9669|1.9114|
> |$\lambda_{zero}=0$|0.2952|0.7792|0.6196|0.3780|0.5933|
> |**all(paper)**|**0.3041**|**0.7854**|**0.6216**|**0.1183**|**0.3439**|
>
> **Revision in updated manuscript.** We will add this table and discussion to the experimental section as a dedicated ablation on w/o relevance / w/o preservation / w/o span / w/o zero.
>
> ### W2. “Minimal and sufficient” claim
>
> We agree that the phrase “minimal and sufficient” is too strong if interpreted as a strict global optimality claim. Our theory supports a density-threshold greedy rule for a monotone submodular objective, which yields an approximation-guided and task-adaptive compact subset, but does not prove strict global minimality or formal sufficiency for a highly nonlinear diffusion model.
>
> What we intended to claim is an empirically compact yet effective subset. This is supported by:
> (i) the diminishing-return behavior in Fig. 3 and the analysis in Sec. 5.5, where cost-aware density stops earlier than gain-only selection and yields a smaller subset; and
> (ii) the ablation against fixed-budget Top-K without cost-aware stopping, where the adaptive strategy achieves stronger alignment and lower forgetting, supporting the empirical sufficiency of this compact subset.
>
> **Revision in updated manuscript.** We will revise this wording throughout the paper to a more precise phrasing such as “task-adaptive compact and empirically sufficient subset” or “approximation-guided compact subset.”
>
> ### W3. Personalization success, expression generalization, and interference
>
> First, regarding generalization to different expressions of the same concept, our current evaluation already goes beyond a single canonical prompt. As stated in Appendix C.1, each concept is evaluated using 20 diverse prompts, ranging from simple re-contextualization to more compositional variants. Thus, the reported CLIP/DINO results already reflect generalization beyond prompt memorization.
>
> Second, regarding interference with other concepts, the current paper already evaluates it from both the output space and the parameter space: Table 2 / Fig. 2 quantify forgetting through TFR and concept-wise decline curves, while Fig. 5 analyzes cross-concept neuron overlap. Together, these results show that SCNS reduces both performance-level interference and parameter collision relative to ablations and continual baselines.
>
> Third, we agree that the current manuscript emphasizes aggregate end-of-stream results more than per-concept evolution. We therefore add the following finer-grained concept-level statistics across stages:
>
> |model|sample|clip_t|clip_i|dino_i|
> |---|---|---|---|---|
> |cat|cat|0.2826|0.8574|0.7140|
> |dog|cat|0.2934|0.8606|0.7173|
> |dog|dog|0.2855|0.8320|0.7075|
> |barn|cat|0.2851|0.8567|0.7131|
> |barn|dog|0.2972|0.8290|0.7128|
> |barn|barn|0.2963|0.7745|0.6515|
> |teddybear|cat|0.2810|0.8635|0.7067|
> |teddybear|dog|0.2913|0.8360|0.7119|
> |teddybear|barn|0.2966|0.7700|0.6561|
> |teddybear|teddybear|0.3250|0.8122|0.5795|
> |vase|cat|0.2809|0.8635|0.7159|
> |vase|dog|0.2995|0.8314|0.7020|
> |vase|barn|0.2986|0.7670|0.6478|
> |vase|teddybear|0.3245|0.8154|0.5831|
> |vase|vase|0.3142|0.7905|0.6055|
> |robot|cat|0.2809|0.8606|0.7139|
> |robot|dog|0.2977|0.8260|0.7057|
> |robot|barn|0.2973|0.7653|0.6354|
> |robot|teddybear|0.3213|0.8128|0.5882|
> |robot|vase|0.3102|0.7944|0.6059|
> |robot|robot|0.3069|0.7910|0.6065|
> |Cartoon|cat|0.2878|0.8610|0.7227|
> |Cartoon|dog|0.2931|0.8340|0.7049|
> |Cartoon|barn|0.2948|0.7750|0.6367|
> |Cartoon|teddybear|0.3199|0.8157|0.5920|
> |Cartoon|vase|0.3095|0.7890|0.6095|
> |Cartoon|robot|0.3036|0.7940|0.6098|
> |Cartoon|Cartoon|0.2887|0.7116|0.5854|
> |Vangogh|cat|0.2890|0.8570|0.7163|
> |Vangogh|dog|0.2935|0.8305|0.7034|
> |Vangogh|barn|0.2996|0.7730|0.6474|
> |Vangogh|teddybear|0.3191|0.8130|0.5809|
> |Vangogh|vase|0.3110|0.7895|0.6023|
> |Vangogh|robot|0.3016|0.7914|0.6090|
> |Vangogh|Cartoon|0.3014|0.7070|0.5792|
> |Vangogh|Vangogh|0.3187|0.7280|0.5565|
>
> **Revision in updated manuscript.** We will include the full 20 evaluation prompts for animal, object, and style concepts in Appendix C.2. We will also add the fine-grained stage-wise table above in the appendix to report concept-level personalization more explicitly.

---

> > ### Author Rebuttal · Reviewer_1MAP · 2026-04-03
> >
> > I thank the authors for their responses. Most of my concerns have been addressed. However, I would strongly suggest avoiding the term “compact.” It still implies a restrictive condition and may lead to overclaim without sufficient justification.

---

> > > ### Author Response · Authors · 2026-04-03
> > >
> > > Thank you for the helpful follow-up and for pointing this out. We agree that the term “compact” may still imply a stronger claim than we intend to make. In the revision, we will avoid this wording and replace it with a more neutral expression, so that the claim better matches our current theoretical and empirical support.

---

### Official Review · Reviewer_4roP · 2026-03-12

**Soundness:** 3
**Presentation:** 2
**Significance:** 3
**Originality:** 3
**Overall Recommendation:** 4
**Confidence:** 3

**Summary:**

This paper addresses the critical challenges of catastrophic forgetting and parameter inefficiency in the Continual Personalization of Diffusion Models. Existing approaches often assume static concepts or rely on computationally expensive model merging, rendering them suboptimal for scenarios where concepts arrive sequentially and replaying old data is infeasible due to privacy or storage constraints.

**Compliance With Llm Reviewing Policy:**

Affirmed.

**Key Questions For Authors:**

1. The method utilizes 100 manually constructed "universal semantic anchors" to compute task relevance. How does the SNR calculation perform when encountering extremely rare or domain-specific novel concepts that fall outside this universal distribution? Is there a mechanism for automatic anchor construction?
2. Experiments focus on sequences of 8–10 concepts. As the sequence length extends to 50 or 100 concepts, while Fisher cache growth is linear, will the space of "available fresh parameters" become exhausted? In such saturation scenarios, would the adaptive stopping criterion select highly overlapping neurons, leading to performance collapse?

**Limitations:**

No. The limitations discussion is not sufficient. The paper should explicitly acknowledge that:
The method does not address conflicts arising during multi-concept simultaneous generation, limiting its utility in complex creative workflows.

**Strengths And Weaknesses:**

STRENGTH
• The paper elevates parameter selection from heuristic ranking to a rigorous Submodular Optimization framework. Explicitly modeling and suppressing semantic redundancy via the "diminishing returns" property is a novel and theoretically sound perspective that addresses the neglect of collective coverage in existing methods.
• Experimental results indicate that SCNS significantly reduces the Task Forgetting Rate (TFR) compared to baselines. This validates the efficacy of combining Fisher-weighted risk and submodular span in balancing stability and plasticity.
WEAKNESS
• The authors acknowledge that the method focuses primarily on Continual Acquisition rather than optimizing for Combinatorial Composition. In real-world applications, users often require the simultaneous generation of multiple learned concepts; SCNS's performance in such scenarios remains unexplored.
• The reliance on diagonal Fisher Information approximation may overlook parameter correlations inherent in deep neural networks. While empirical results are strong, this theoretical simplification could potentially lead to misidentification of critical parameters.

---

> ### Author Rebuttal · Authors · 2026-03-30
>
> We thank the reviewer for the positive assessment of the technical novelty and empirical effectiveness of SCNS. Below we address each concern and clarify the corresponding revisions in the updated manuscript.
>
> ### W1. Multi-concept simultaneous generation / composition & Limitations discussion
>
> Thank you for highlighting this important scope issue. Our method is designed for continual acquisition: learning a new personalized concept while preserving previously acquired ones through parameter-space allocation and protection. It does not directly resolve inference-time conflicts in multi-concept simultaneous generation. We agree this boundary should be stated more explicitly.
>
> **Revision in updated manuscript.** We will move this limitation from the Appendix E to the main paper and make it explicit in the Conclusion/Limitations section, while also revising the Abstract/Introduction wording to avoid suggesting that multi-concept simultaneous composition has been validated. We will further state that SCNS is complementary to inference-side composition/control techniques rather than a replacement for them.  In Future Work, we will briefly discuss combining SCNS with downstream composition/control modules for multi-concept generation.
>
> ### W2. Diagonal Fisher approximation
>
> We agree that a diagonal Fisher approximation does not capture full inter-parameter correlations. In SCNS, however, Fisher is used in a restricted role: it appears only in the preservation-risk term as a lightweight second-order sensitivity proxy for each candidate neuron, rather than as a full model of parameter interactions. Its purpose is to provide a conservative estimate of which units are risky to perturb under the strict replay-free setting. More generally, diagonal Fisher is widely used as an efficient surrogate for full Fisher, with much lower cost and strong empirical performance.
>
> **Revision in updated manuscript.** We will revise Sec. 4.3.2 and the corresponding appendix discussion to state clearly that Fisher is used as a local risk proxy for preservation, not as a claim of modeling the full covariance structure. We will also strengthen the limitation statement to note that richer structured approximations are a meaningful future direction.
>
> ### Q1. Universal semantic anchors and rare/domain-specific concepts
>
> The 100 universal semantic anchors are not intended to enumerate future concepts. Their role is to estimate a neuron’s general responsiveness so that the relevance score favors units that respond strongly to the current concept but not broadly to generic semantics. Concretely, the anchors appear in the denominator of the SNR-style relevance score, while the numerator is the gradient response to the current concept. Therefore, for rare or domain-specific concepts, the method does not require the concept itself to appear in the anchor set; it only requires that concept-specific units have large current-task response relative to broadly active “generalist” units.
>
> **Revision in updated manuscript.** We will clarify this point in Sec. 4.3.1 and Appendix C.3, explicitly stating that anchors serve as a general-response baseline, not a coverage set over all future concepts. We will also add a short paragraph discussing that automatic anchor construction is a promising extension, but is not required for the present mechanism.
>
> ### Q2. What happens for 50/100 concepts? Will fresh parameters be exhausted?
>
> This is an important question. As discussed in Appendix B.2, the cost-aware density term acts as a dynamic capacity reservation mechanism: neurons with repeated historical usage incur higher cost, which steers selection away from saturated regions, and the adaptive stopping rule terminates once no candidate retains sufficient utility-to-cost ratio.
>
> This is consistent with the empirical results. The extended-stream analysis shows smoother degradation for full SCNS than for the score-only variant, and Appendix D.4 / Table 6 show that the parameter allocation ratio gradually decreases as more concepts are learned. Together, these results suggest that under very long concept streams, SCNS is more likely to fail from insufficient free capacity for later concepts than from aggressively selecting highly overlapping neurons that would immediately damage earlier ones.
>
> **Revision in updated manuscript.** We will make this long-stream failure mode explicit in the Limitations/Future Work discussion: for extremely long streams, SCNS prioritizes retention by avoiding destructive overlap, so the main bottleneck becomes insufficient free capacity for later concepts. In Future Work, We will also briefly note that parameter expansion or other capacity-growth strategies are natural extensions once such capacity shortage is detected.

---

### Decision · Program_Chairs · 2026-04-30

**Decision:**

Accept (regular)

**Comment:**

The paper proposes a method for teaching a diffusion model new personalized concepts one at a time without forgetting old ones.
Instead of updating the full model, it selects an optimal subset of neurons, considering the joint effect of different neuron combinations on the target objective.

Scores - 4 (WA), 4 (WA), 4 (WA)

Reviewers find the approach novel, well-motivated, and effective.
Concerns include overclaiming,
missing ablations per metric, and that the method only handles learning concepts one-by-one, without the ability to generate multiple learned concepts together in one image.

In the rebuttal, the authors provided a component ablation showing each metric contributes, softened the overclaims, and added fine-grained per-concept evaluation results.

During the discussion phase, it was noted that a simpler baseline - training a separate adapter per concept and loading the relevant one at inference - might sidestep the need for continual learning entirely. 1MAP argued that single-model continual acquisition is a meaningful contribution on its own.

The Area Chair leans towards acceptance. The camera-ready would benefit from reflecting the rebuttal discussions and discussing extension to multi-concept composition in the main paper.